# Structure mapping of dengue and Zika viruses reveals functional long-range interactions

Roland G. Huber [1], Xin Ni Lim[2], Wy Ching Ng[3], Adelene Y.L. Sim[1], Hui Xian Poh[2], Yang Shen[4], Su Ying Lim[2], Karin B. Sundstrom[3], Xuyang Sun[5,6], Jong Ghut Aw[2], Horng Khit Too[7,8], Peng Hee Boey[7,8], Andreas Wilm[4], Tanu Chawla[3], Milly M. Choy[9,10], Lu Jiang[11,12], Paola Florez de Sessions[13], Xian Jun Loh[11,12], Sylvie Alonso[7,8], Martin Hibberd[9,10], Niranjan Nagarajan[4], Eng Eong Ooi [3,7,14], Peter J. Bond [1], October M. Sessions[3,14,15] & Yue Wan[2,16,17]

Dengue (DENV) and Zika (ZIKV) viruses are clinically important members of the Flaviviridae family with an 11 kb positive strand RNA genome that folds to enable virus function. Here, we perform structure and interaction mapping on four DENV and ZIKV strains inside virions and in infected cells. Comparative analysis of SHAPE reactivities across serotypes nominates potentially functional regions that are highly structured, conserved, and contain low synonymous mutation rates. Interaction mapping by SPLASH identifies many pair-wise interactions, 40% of which form alternative structures, suggesting extensive structural heterogeneity. Analysis of shared interactions between serotypes reveals a conserved macro-organization whereby interactions can be preserved at physical locations beyond sequence identities. We further observe that longer-range interactions are preferentially disrupted inside cells, and show the importance of new interactions in virus fitness. These findings deepen our understanding of Flavivirus genome organization and serve as a resource for designing therapeutics in targeting RNA viruses.

[1] Bioinformatics Institute (A*STAR), 30 Biopolis Street #07-01, Matrix, Singapore 138671, Singapore. [2] Stem Cell and Regenerative Biology, Genome Institute of Singapore, Singapore 138672, Singapore. [3] Program in Emerging Infectious Diseases, Duke-NUS Graduate Medical School, 8 College Road, Singapore 169857, Singapore. [4] Computational Biology, Genome Institute of Singapore, Singapore 138672, Singapore. [5] Department of Neurology, National Neuroscience Institute, 20 College Road, Singapore 169856, Singapore. [6] Stem Cell and Regenerative Biology, Genome Institute of Singapore, 60 Biopolis Street, Singapore 138672, Singapore. [7] Department of Microbiology and Immunology, Yong Loo Lin School of Medicine, National University of Singapore, Singapore 117545, Singapore. [8] Immunology programme, Life Sciences Institute, National University of Singapore, Singapore 119077, Singapore. [9] Infectious Diseases, Genome Institute of Singapore, Singapore 138672, Singapore. [10] London School of Hygiene & Tropical Medicine, London WC1E 7HT, UK. [11] Institute of Materials Research and Engineering (IMRE), A*STAR, Singapore 138634, Singapore. [12] Department of Materials Science and Engineering, National University of Singapore, Singapore 117575, Singapore. [13] GERMS Platform, Genome Institute of Singapore, Singapore 138672, Singapore. [14] Saw Swee Hock School of Public Health, National University of Singapore, Singapore 117549, Singapore. [15] Department of Pharmacy, National University of Singapore, Singapore 117559, Singapore. [16] School of Biological Sciences, Nanyang Technological University, Singapore 637551, Singapore. [17] Department of Biochemistry, Yong Loo Lin School of Medicine, National University of Singapore, Singapore 117597, Singapore. These authors contributed equally: Roland G. Huber, Xin Ni Lim, Wy Ching Ng, Adelene Y. L. Sim. Correspondence and requests for materials should be addressed to P.J.B. (email: peterjb@bii.a-star.edu.sg) or to O.M.S. (email: october.sessions@duke-nus.edu.sg) or to Y.W. (email: wany@gis.a-star.edu.sg)

Dengue (DENV) and Zika (ZIKV) viruses are members of the flavivirus genus of the Flaviviridae family of RNA viruses and are important human pathogens imposing a high economic and social burden worldwide[1]. DENV is predicted to infect 390 million people per year, resulting in dengue fever, and in severe cases, death[1]. The Zika outbreak in Brazil was declared a public health emergency of international concern by the World Health Organisation in 2016 and has been associated with microcephaly in neonates and Guillain-Barre syndrome in adults[2].

The genome of DENV and ZIKV viruses consists of an ~11 kb-long positive strand RNA that encodes a single polyprotein that is then post-translationally cleaved into ten viral proteins consisting of three structural proteins (C, prM, and E) and seven non-structural (NS) proteins (NA1, NS2A, NS2B, NS3, NS4A, NS4B and NS5)[3]. In addition to the primary sequence of these genomes, a better understanding of how the genome is structurally organized is important for understanding virus function[4–8]. Highly structured elements and long-range interactions in the 5′ and 3′ terminal regions, including the capsid region, of flaviviral genomes have been shown to be important for translation and replication of these viruses[9–12]. Many of these structural features are present across the flavivirus family, indicating a high degree of selection pressure to retain them. Additional local structures throughout the genome have been computationally predicted to exist by Proutski et al.[13]. However, these predictions and their potential functional relevance to the viral life cycle have not been fully assessed.

Here we perform genome-wide RNA secondary structure and interactome mapping on all four serotypes of DENV (DENV1–4) and four geographically distinct ZIKV viruses (African, Brazilian, French Polynesian, and Singaporean)—representing the known genetic diversity of this emergent virus. To circumvent limitations in RNA structure probing of virus RNAs in vitro, related to changes in solvent conditions (i.e. altered RNA–protein interactions and the absence of the virus envelope[14,15]), we perform structure probing of DENV and ZIKV inside their native virus particles and in human host cells (Fig. 1a). We observe that these genomes are highly structured and identify conserved structures across these viruses. In addition to the known circularization signal, we show that many additional long pair-wise interactions exist and are important for virus fitness. Finally, comparison of in cell and in virion RNA interactions show that many interactions are disrupted in the cytoplasm, suggesting that they may be actively unwound by helicases inside the cell.

## Results

**2-Methylnicotinic acid imidazolide-mutational mapping of DENV and ZIKV genomes inside virus particles.** To map secondary structures across the viral genome, we treated intact virus particles with a SHAPE-like chemical, 2-methylnicotinic acid imidazolide (NAI)[16], which has been shown to modify single-stranded regions along RNAs efficiently, in vivo. We then extracted the RNA genomes from the virus particles and identified the modification sites by mutational mapping (MaP) (Fig. 1a)[17]. We validated that NAI-MaP signals are accurate in vivo, based on the 28S rRNA of Hela cells (Supplementary Figure 1a).

We performed at least two biological replicates of NAI-MaP on each of the four DENV and four ZIKV viruses, and sequenced >120 million reads per sample (Supplementary Table 1). This resulted in >100 reads mapped per base on 99.99% of all bases across the eight viruses, yielding structure information for most of the bases along each genome (Fig. 1b, Supplementary Data 1). The NAI-MaP reads between two biological replicates were well correlated to each other (Pearson correlation, $p > 0.81$),

suggesting that the structure probing was of good quality. Gel electrophoresis of extracted DENV and ZIKV RNA showed that the majority of the genomic RNA is intact (Supplementary Figure 2a, b), indicating that structure probing was performed while the RNA was in its full-length context. We observed good correlation between our NAI-MaP reactivities and known structures in the 5′ and 3′ untranslated region (UTR) of DENV and ZIKV genomes, with 68–94% of high-reactivity bases falling in known single-stranded regions in DENV and ZIKV (Fig. 1c, Supplementary Figure 2c, d)[10,18,19], indicating that our structure data are of good quality. We also checked that our NAI-MaP signals agree with an orthogonal high-throughput enzymatic structure probing method (parallel analysis of RNA structures) on renatured DENV1 RNA[20] (Supplementary Figure 3a, b), and that the distribution of NAI-MaP reactivities agree with known paired and unpaired bases in literature (Supplementary Figure 3c)[10]. In addition, we also calculated Shannon entropies[17], along the entire length of DENV and ZIKV genomes, to identify regions that are likely to form unique structures (Fig. 1b, bottom).

Structure probing across different viruses allows us to perform comparative structure analysis across the eight viruses to identify shared structures. As the four DENV serotypes share around 60–70% sequence identities with each other and ~58% sequence identity with the ZIKV strains (Supplementary Figure 4a), the sequence divergence allows us to analyze the relationship between their structure conservation and sequence identity. We observed that unpaired bases shared across the viruses tend to have higher sequence conservation (Supplementary Figure 4b), suggesting that these regions might contain important sequence information for gene regulation such as for interaction with RNA-binding proteins. While regions that share sequence similarity tend to share structural similarity (Supplementary Figure 4c, Fig. 1d), important structural elements under evolutionary constraints could show a greater similarity in structure over sequence across the different viruses. To test this, we calculated regions that show significantly higher structural similarity over sequence similarity and found surprisingly consistent regions, including the NS2A region, across the four different DENV strains, suggesting that these regions are under structure conservation across the viruses (Fig. 1d, starred).

To further identify potentially important structures along the genome, we searched for regions that: (1) are structurally similar; (2) are highly structured; and (3) show low synonymous mutation rates[21]. Synonymous mutation rates were calculated along hundreds/thousands of DENV sequences and hundreds of ZIKV sequences within each serotype, to identify mutational cold spots within the genome (Supplementary Figure 5a, b, Methods). We nominated 16 DENV and 12 ZIKV consensus RNA regions that fulfilled two of three criteria, namely that they: (i) are highly structured; (ii) share similar NAI reactivity patterns; and (iii) show low mutation rates within and across serotypes (Fig. 2a, Supplementary Figure 5c, in purple, Methods). These regions overlap with RNA elements that show evolutionary pressure at the structural level (see above), further confirming their importance. As many of these consensus regions have significantly lower Shannon entropies than average, indicating they are likely to form unique structures (Supplementary Figure 5d), we modeled their secondary structures using our NAI-MaP reactivities as structural constraints.

To model the RNA structures present within the DENV and ZIKV genomes, we incorporated our NAI-MaP reactivities into the RNAstructure suite of programs to generate an ensemble of potential structures (Methods)[21], and selected structures that mapped most accurately to the known 5′ and 3′ UTRs of DENV and ZIKV virus (Fig. 2b, Supplementary Figure 6,). The modeled 5′ and 3′ UTR structures of the 4 DENV and ZIKV viruses have

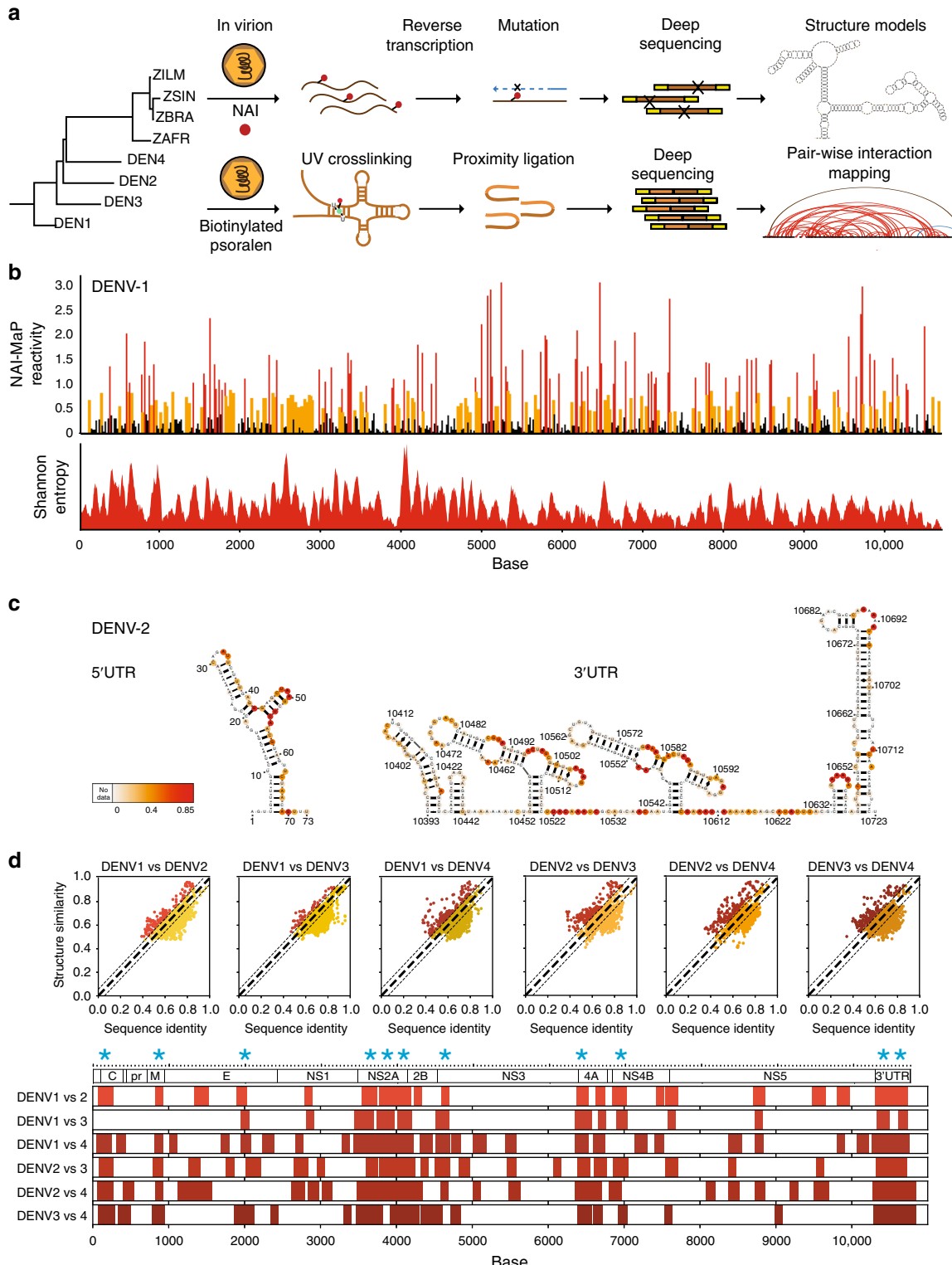

an accuracy of 91 and 87% (DENV1), 92 and 93% (DENV2), 78 and 86% (DENV3), 91 and 63% (DENV4), 76 and 73% (ZIKV African), 81 and 84% (ZIKV Brazil), 78 and 78% (ZIKV French Polynesia), and 85 and 83% (ZIKV Singapore) for 5′ and 3′ UTR structures respectively (Supplementary Figure 7). We verified that incorporating NAI-MaP reactivities into structure modeling significantly improved the accuracy of the structure models using 16S and 23S rRNA (Supplementary Figure 8a, Methods). Structure models incorporating NAI-MaP data are enriched for

covaried bases, supporting the existence of these structures inside virus particles (418 and 430 covaried bases in modeled DENV1 and ZIKV French Polynesia structures versus 258 and 276 in shuffled controls, Methods)[22]. Bases with NAI-MaP reactivities that do not agree with the modeled structure are enriched for high Shannon entropies, suggesting that they may form alternative conformations (Supplementary Figure 8b). Interestingly, similarly to other flaviviruses such as Hepatitis C Virus (HCV)[8], all of the eight DENV and ZIKV genomes maintain a

**Fig. 1** Genome-wide structure mapping of DENV and ZIKV genomes inside virions. **a** Schematic showing the workflow for identifying functional structural elements in eight viruses. Full-length virus genomes are probed inside their virus particles using a SHAPE-like chemical, NAI, which modifies single-stranded regions along the genome. Pair-wise interactions within the virus genomes are also interrogated using biotinylated psoralen, which crosslinks base-paired regions inside the virus. The local and pair-wise experimental data are then used to constrain computational models to derive more accurate structure predictions for DENV and ZIKV. We also identify structurally conserved virus structures and determine the functional impacts of conserved and pair-wise interactions through mutagenesis and virus fitness assays. **b** NAI-MaP reactivities and Shannon entropy along viral genome. Top: raw NAI-MaP reactivities along DENV1: Y-axis indicates the extent of NAI reactivity, and X-axis indicates position along the genome. Red, yellow, and black bars indicates high, medium, and low reactivities respectively. Bottom: Shannon entropy across the DENV1 genome. Low values indicate higher probability of having a defined structure. **c, d** NAI-MaP structure signals maps to known structures in the 5′ UTR and 3′ UTR of DENV2 virus genomes. A higher NAI-MaP reactivity is colored to be more red, indicating that a base is likely to be single-stranded. **d** Top: scatterplot showing the sequence identity and structure similarity derived by NAI-MaP between each pair of DENV strains in 100-nt windows. The red dots show locations whereby structure similarity is greater than sequence similar by 1 standard deviation from mean. Bottom: each row indicates the locations of the high structure/sequence similarity regions (top) in each DENV pair, along the DENV genome. The structure conserved regions are consistently located at specific locations along the genome (200-, 900-, 2000-, 3800-, 4000-, 4200-, 4600-, 6500-, 7000-, 10,400-, and 10,600-nt regions, starred), indicating that the high structure conservation is not random and is an evolutionary constraint on structure beyond constraints on sequence alone

---

short median helix length of four consecutive canonical base pairs, with 85–90% of the base pairs existing in helices that contain seven consecutive base pairs or less, which is likely to be important for the genomes to evade immune surveillance inside cells (Supplementary Figure 8c, d).

**Long-range pair-wise interactions are abundant in DENV and ZIKV genomes.** Beyond local RNA base pairing, long-range RNA–RNA interactions have been shown to play important roles in viral replication by enabling the genome to circularize and position the RNA-dependent RNA polymerase close to the transcription start site[9,11,23]. The three principal RNA sequences of the DENV genome responsible for the long-range interaction of the 5′ and 3′ ends are: (i) the circularization sequence (CS); (ii) the upstream AUG region (UAR); and (iii) the downstream AUG region (DAR)[23]. A fourth interaction region located at 150 bases (C1 structure of the capsid protein) and the dumbbell of the 3′ UTR have also been found to be involved in genome circularization[11]. To directly capture long-range pair-wise interactions that span distances longer than 500 bases in DENV and ZIKV genomes comprehensively, we performed two biological replicates of sequencing of psoralen-crosslinked, ligated, and selected hybrids (SPLASH) on each of the eight viral genomes, inside their viral particles, using biotinylated psoralen, proximity ligation, and deep sequencing (Fig. 1a)[24]. We further enriched our data for true interactions by filtering against random interactions that could occur by permutation (Supplementary Figure 9a).

The resultant SPLASH data revealed thousands of intramolecular pair-wise RNA interactions inside virus particles (median distance of interaction = 3.6 kb), greatly expanding the list of long-range interactions known for any family of RNA viruses (Supplementary Data 2, 3). SPLASH interactions are enriched for covaried bases, supporting the importance of their existence in DENV and ZIKV genomes (Supplementary Figure 9b). We observed interactions that arise from the known CSs in both the DENV and ZIKV viruses as one of our top interactions (Fig. 3a), indicating that genome circularization is not only important in vivo but is also maintained in virions. To increase the resolution for identifying interaction positions, we performed peak calling of the mapped reads on each end of the interaction (Methods). Peak calling revealed clear peaks that were around 50 bases wide, centered at the known 5′ and 3′ UAR and DAR interaction sites (Fig. 3b). Hybridization of the 5′ and 3′ peak regions using the program RNAcofold revealed the interaction sites of the UAR, DAR, and CS regions in the DENV genomes (Fig. 3b)[25], demonstrating the precision of the SPLASH data.

**DENV and ZIKV genomes form an extensive ensemble of alternative structures.** Previous studies in HCV suggest that its genome can fold into alternative conformations[7]. To determine the amount of structural heterogeneity in the DENV and ZIKV genomes, we calculated the number of RNA regions that are observed to interact with two or more other regions along the genome. We observed that 48.8% of long (≥500 bases) and 41.6% of short (<500 bases) pair-wise interactions fold into alternative structures, indicating a large amount of structural heterogeneity within the genomes (Fig. 3c, Supplementary Figure 9c). Interestingly, the alternative interaction sites are enriched in regions with significantly higher Shannon entropies, confirming that these regions tend to take on more than one structure (Fig. 3d). To test whether the different alternative structures are generally functional or whether only the most stable interaction is important to the virus, we mutated long-range interactions that could pair with three different regions along the ZIKV French Polynesia genome (1786:5290, 1784:7034, and 1784:8462), disrupting the interactions that bring the nucleotide sequences encoding the envelope (E) protein into close proximity with those of NS3, NS4B, and NS5, respectively (Fig. 3e, Supplementary Figure 9d). As we chose mutations that disrupt pair-wise interactions and preserve the amino-acid sequence and codon frequencies, we were unable to design for compensatory mutations (Fig. 3f). Mutations along each individual strand in the different alternative structures resulted in a decreased amount of virus being produced in both infected Huh7 liver cells and the supernatant (Fig. 3g), suggesting that the mutations attenuated virus fitness.

**SPLASH reveals two distinct modes of conserved long-range interactions.** Interactome mapping across the four DENV and four ZIKV strains allowed us to identify shared pair-wise interactions that are present across different viruses (Supplementary Data 4, 5). Forty-nine percent of the top DENV interactions are shared between at least two DENV serotypes, and 78% of the top ZIKV interactions are shared between at least two ZIKV serotypes (Fig. 4a, c). We observed that short-range interactions are more likely to be shared than long-range interactions, suggesting that they are more conserved (Supplementary Figure 10a). The large number of shared pair-wise interactions between the ZIKV strains (in particular between the Brazil, French Polynesia, and Singapore strains) reflect the high similarity in sequence between the ZIKV viruses and the robustness of our SPLASH strategy, suggesting that the genomes form consistent folds inside virions (Fig. 4c).

We observed that there are two types of pair-wise interactions that maintain the overall topology of these viruses. The first

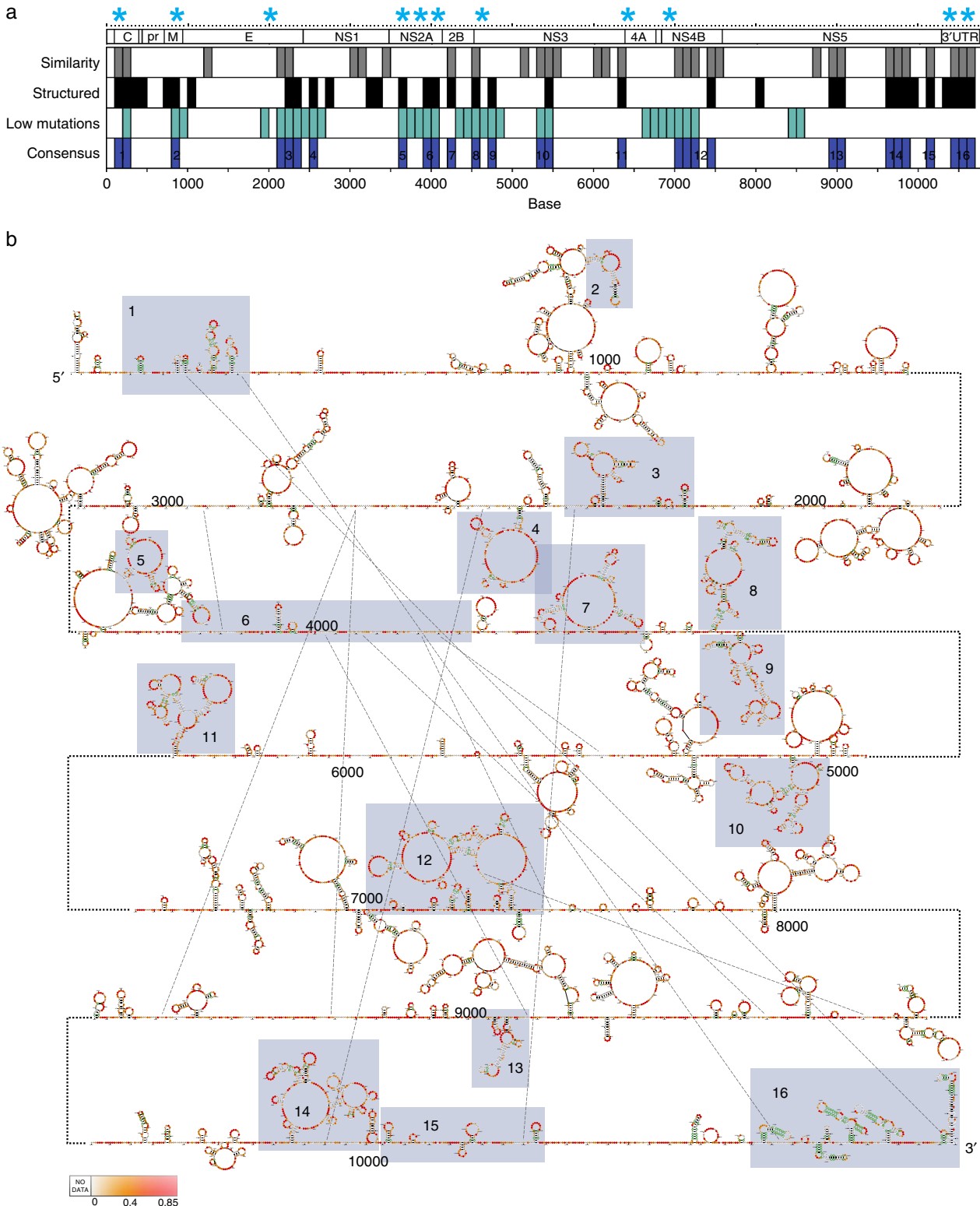

**Fig. 2** DENV genome contains many conserved structural elements. **a** Plots show 100-nucleotide regions across four DENV genomes that have highly similar structures (gray), are highly double-stranded (black), and accumulate low levels of synonymous mutations (blue). Regions that fulfill two out of three of the above criteria are selected as consensus regions (purple) and are potentially functionally important. The structurally conserved regions from Fig. 1e are starred in blue. **b** A structure model of the DENV1 genome using NAI-MaP as experimental constraints, for the 16 consensus regions identified in **a**. The NAI-MaP reactivities and covariation information (green circles) are mapped onto the structures

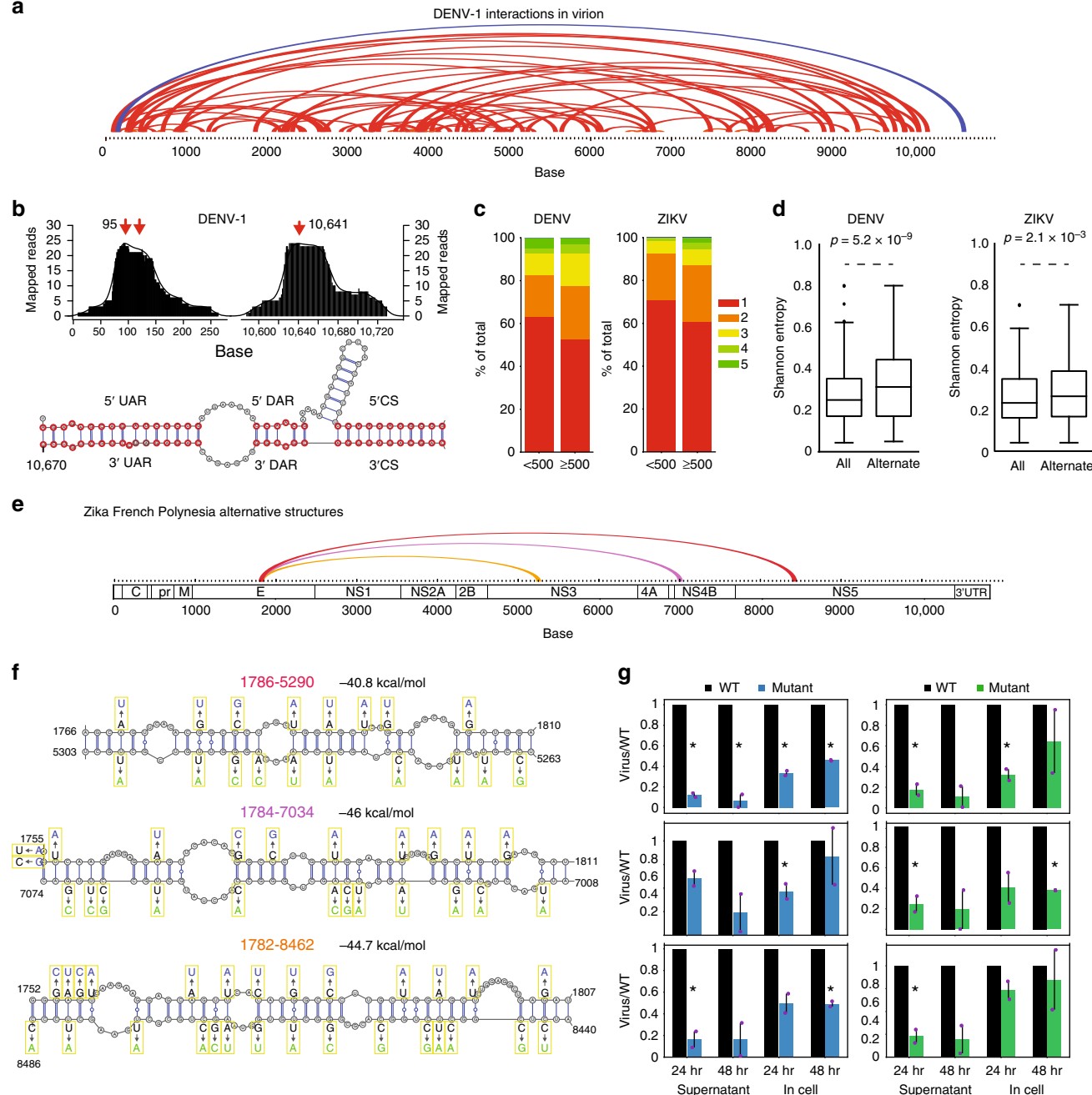

**Fig. 3** DENV and ZIKV genomes form extensive pair-wise interactions. **a** Arc plot showing the most abundant top 75 pair-wise interactions in the DENV1 genome inside virions. The known circularization signal is colored in dark blue. **b** Top: peak calling of the circularization interaction region shows that the peak of base pairing occurs at positions 95 and 10,641. Bottom: prediction of hybridization using RNAcofold program using sequences around the called peaks identified three of the best known circularization signals. **c** Bar charts showing the number of peaks that interact uniquely and with other regions along the genome. Greater than forty percent of DENV and ZIKV interactions are not unique. **d** Boxplot showing that regions with alternative base pairing (alternate) have higher Shannon entropies than all regions present in the genome (all) for both DENV1 and ZIKV French Polynesia strain. Box encompasses first and third quartiles, with the median indicated in the box. Whiskers are at minimum and maximum of points excluding deviations of 1.5× interquartile (Q3–Q1) range from Q1 to Q3 as outliers. Outliers are indicated as dots. *p*-Value calculated using *T*-test. **e** Arc plot showing one set of alternative interactions found along the ZIKV French Polynesia genome. **f** Structure models of the pair-wise interactions for the alternative structures in **e**, using the program RNAcofold. Mutations along each strand of the pair-wise interaction are in blue and green respectively. **g** qPCR analysis of the amount of mutant (from **f**) and WT virus that are produced in Huh7 cells and supernatant, 24 and 48 h post infection. The *Y*-axis shows that amount of mutant virus produced as compared to wild type. Statistical analysis was performed using Student's *T*-test (one tail) to determine if the mutant viruses are attenuated compared to wild type. **p*-value < 0.05. Error bars represent SEM. Two biological replicates of Gibson assembly were performed to generate WT and MT viruses and two biological replicates of Huh7 infection were performed

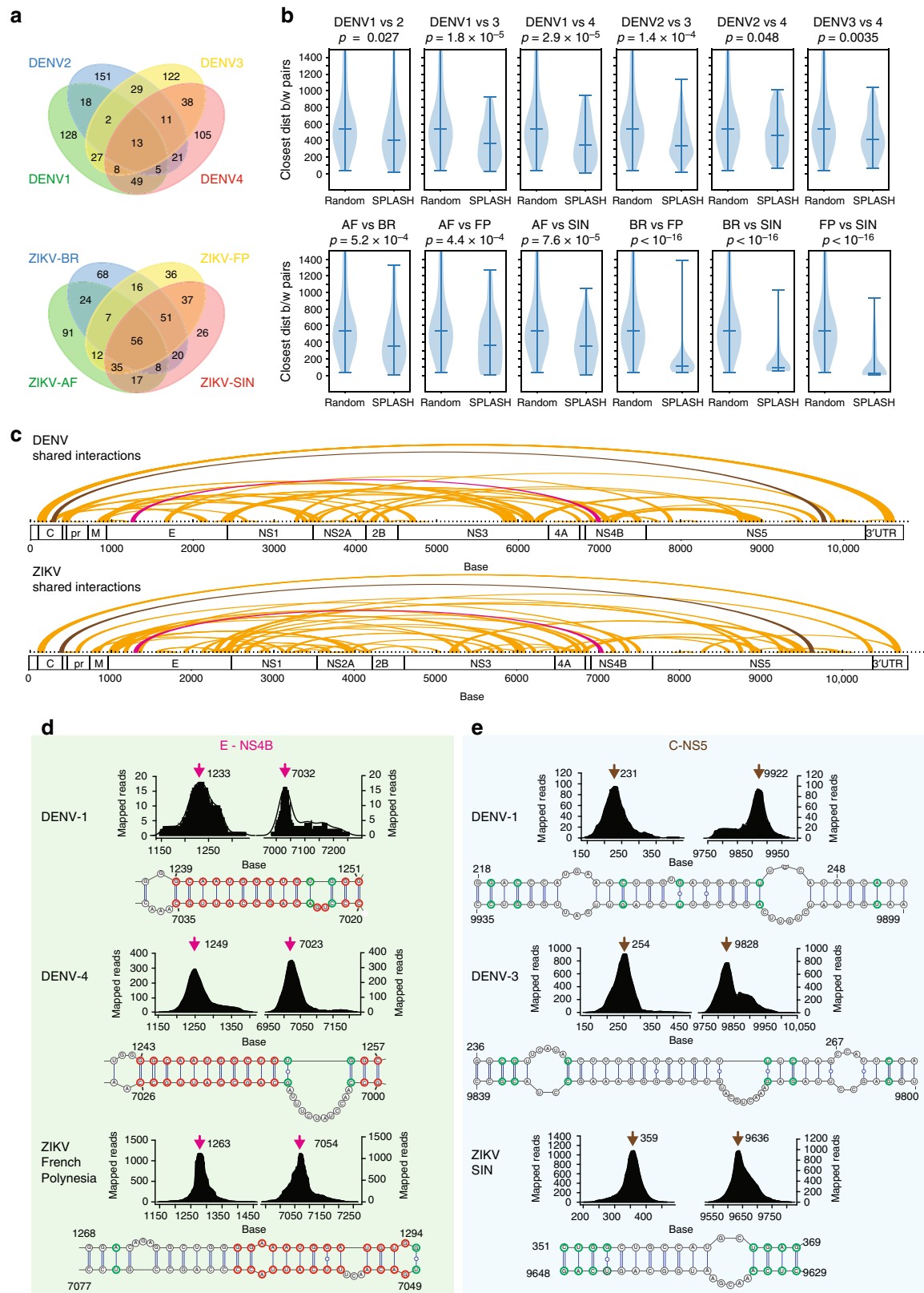

interaction type preserves the exact spatial locations as well as the sequences of the interaction across serotypes. This is exemplified by the discovery of a conserved long-range interaction between E and NS4B (~bases 1200:7000) in DENV and ZIKV—whereby different viruses use homologous sequences to form the long-range interaction (Fig. 4d). We also observed a second type of interaction, whereby viruses utilize different sequences in nearby spatial locations to bring two distant regions of the genome into close proximity. We observed that stable pair-wise interactions present in one virus serotype frequently have a similarly shared interaction that is present nearby in another serotype—suggesting flexibility in utilizing proximal sequences to achieve overall

**Fig. 4** Two different modes of long-range interactions in DENV and ZIKV. **a** Venn diagram showing overlap of DENV (top) and ZIKV (bottom) interactions between two, three, and four viral strains, for the 250 most prominent interactions in each strain. The interaction is considered to be shared if both ends are within ±100 nt of one another (e.g. 1200–7000 would be considered shared with 1299–7000 in another strain). **b** Violin plots showing the distribution of the distance of the closest similar interaction between two viruses versus random. Highly similar viruses such as ZIKV-SIN, ZIKV-FP, and ZIKV-BR show that the average distances to a related interaction are very short, indicating a high level of conservation. *p*-Value is calculated using the Wilcoxon rank-sum test. Minimum, median, and maximum are indicated as horizontal bars. **c** Arc plots showing the top 100 conserved long-range interactions that are present in two of more different DENV (top) and ZIKV viruses (bottom). The pink and gray interactions are further characterized in **d** and **e**. **d** Conserved long-range interaction in DENV and ZIKV viruses that use similar sequences to bring E and NS4B regions together. Top: SPLASH interaction peaks between E and NS4B for different viruses. Bottom: predicted hybridization models for the long-range interaction using RNAcofold. Homologous and covaried bases between the different viruses are circled in red and green respectively. The pink arrows indicate the position of the peak of the interaction. **e** Long-range interaction in DENV and ZIKV viruses that use different sequences to bring NS4 and NS5 regions close together. Top: SPLASH interaction peaks between NS4 and NS5 for different viruses. Bottom: predicted hybridization models for the long-range interaction using RNAcofold. The gray arrows indicate the position of the peak of the interaction

genome packaging (Fig. 4b). These interactions also show significant high covariation over background, suggesting that they are likely to important (Supplementary Figure 10b, Supplementary Data 6,7). Two examples of the second type of interaction exist between the C and NS5 regions (Fig. 4e) or between the NS2A and NS5 regions (Supplementary Figure 10c) in both the DENV and ZIKV genomes. While DENV1 uses bases 218–253 (within C) to pair with bases 9899–9935 (within NS5), DENV3 uses a nearby region—bases 237–274—to pair with another region of 9800–9840 to form a stable structure. In ZIKV viruses, bases 351–369 are used to hybridize with bases 9629–9648 instead. Hypothetically, this second mechanism of genome packaging could enable RNA viruses to tolerate mutations—as long as the overall architecture of the genome is not affected—and could also reflect convergent evolution in different viruses to achieve similar genomic folds.

**Modeling of RNA structure ensembles of the DENV/ZIKV genomes**. As NAI-MaP reactivities and SPLASH data provide different information on how RNAs fold, integrating these two strategies into structure modeling remains a challenge. To integrate both local pairing with pair-wise RNA interaction information into structure modeling, we generated an ensemble of one thousand potential structures using NAI-MaP data as constraints, and subsequently selected for structures that contain the largest numbers of non-conflicting SPLASH interactions (Methods). We confirmed that NAI-MaP constrained, SPLASH-selected structures are significantly more accurate than non-SPLASH-selected structures in 16S and 18S rRNA (*p* < 0.003), suggesting that combining these two sources of different experimental data can improve structure modeling. DENV and ZIKV NAI-MaP constrained, SPLASH-selected structures represent major populations in different structural clusters, and thus reflect the diversity of experimentally observed structures (Supplementary Figures 11–16).

**Longer-range interactions are actively disrupted inside cells**. While structure probing of virus genomes inside their virions provides useful information on their packaging, RNAs could fold very differently inside cells[26,27]. To understand how DENV and ZIKV genomes fold inside infected cells, we performed pair-wise interactome mapping of the four DENV and the four ZIKV viruses in human liver-derived and neuronal precursor cells, respectively, and filtered against random interactions by permutation (Supplementary Figure 17a, Supplementary Data 8, 9). Our interactome data correlate well with another independent high-throughput interactome data set[28], indicating that it is robust and reproducible (*R* = 0.625, Supplementary Figure 17b). Mutational experiments on one side of the pair-wise interactions resulted in

the decreased binding of the other complementary strand, validating the existence of these interactions inside cells (Supplementary Figure 17c). We observed the circularization signal as one of the key longer-range interactions inside cells, further attesting to the quality of our data and the importance of the circularization signal in genome replication[9] (Fig. 5a, Supplementary Figure 18).

Interestingly, we observed that while 77% of pair-wise interactions inside virions are longer than 500 bases, only 34% of pair-wise interactions inside cells are that long (Fig. 5b), suggesting that the genomes are less structured inside cells. Short interactions found inside virions are more likely to be shared inside cells (42% short versus 18% long interactions in DENV virions are shared in cells, and 61% short versus 36% long interactions in ZIKV virions are shared in cells), indicating that they are more stable across various stages of the virus life cycle. We also observed that the majority of the interactions in cells are unique or have only one alternative interaction partner, whereas in the virion, interactions have a higher tendency to form two or more alternative interactions (7.8% interactions with ≥2 interaction partners in cells versus 14.8% in virions, *p* = 0.02), suggesting that genomes are more structurally heterogeneous inside virus particles than in cells. To test whether the genomes are more structured inside virions because of the spatial constraints induced by the virus envelope, and/or whether the genomes are actively being unwound in vivo, we performed interactome mapping on virus genomes that are released from the virions by performing proteinase K treatment on the virus particles (Fig. 5c). Although we observed genome rearrangements when the genome is released in solution—in particular longer-range interactions are disrupted while shorter-range interactions are formed—the genome remains highly structured in solution (Fig. 5c, d). This suggests that the unstructured state we observe inside cells is likely due to active unwinding by cytoplasmic enzymes[26].

Since many of the longer-range interactions in virions are disrupted inside cells, it is unclear whether these pair-wise interactions are functionally important. To determine the significance of the pair-wise interactions that are present in virion, in cell, or in both, we performed mutagenesis experiments to disrupt the interactions found in all three classes (Fig. 5e–i). Disruption of structures that are present in high abundance in any of these three classes resulted in virus attenuation. For one interaction that is abundant in cell and another abundant in virion, we observed less virus production in the supernatant, as compared to inside cells, indicating that they could potentially be involved in genome packaging (Fig. 5e, f). While it is extremely difficult to design compensatory mutations to rescue pair-wise interactions in the virus coding region due to amino-acid and codon frequency constraints, we managed to design point mutations to rescue one of the key interactions that is found

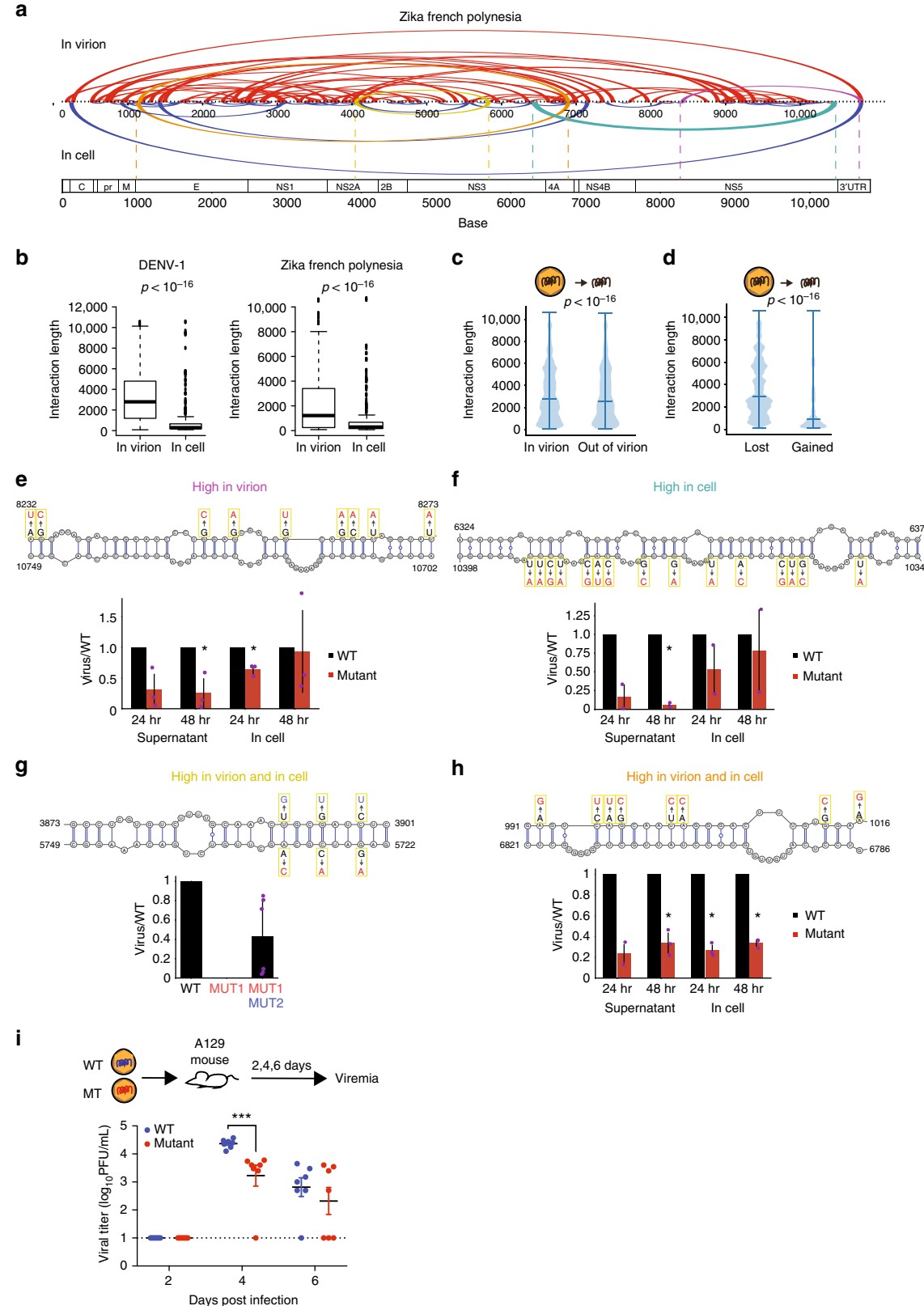

both in cell and in virion (between NS2A and NS3). While mutations in the NS3 region (5723–5750) severely reduced virus replication, this phenotype was partially rescued by designing compensatory mutations in the NS2A region (3871–3901) (Fig. 5g), suggesting that this genomic interaction is important for virus fitness. We also observed virus attenuation in another

set of mutations that disrupt an interaction that is abundant in cell and in virion, disrupting the pairing between E and NS4A coding regions (Fig. 5h). To show that the mutations attenuate virus replication, we injected wild-type and mutant virus into interferon-deficient A129 mice and checked for viremia 2–6 days post infection. Mutant virus resulted in lower virus titers in the

**Fig. 5** Long-range interactions in virions are disrupted inside host cells. **a** Arc plots showing pair-wise interactions identified in virions (top) versus inside host cells (bottom) for the Zika French Polynesia strain. **b** Boxplots showing the distribution of pair-wise interaction lengths inside virions versus inside host cells for DENV1 and ZIKV French Polynesia strains. Box encompasses first and third quartiles, with the median indicated in the box. Whiskers are at minimum and maximum of points excluding deviations of 1.5× interquartile (Q3–Q1) range from Q1 to Q3 as outliers. Outliers are indicated as dots. **c** Violin plot showing the distribution of pair-wise interaction lengths inside virions versus when released into solution by proteinase K treatment. While there is a slight decrease in longer-range interactions, the DENV genome remains highly structured in solution. Minimum, median, and maximum are indicated as horizontal bars. **d** Violin plot showing the length of pair-wise interactions that are disrupted versus interactions that are formed when the DENV genome is released into solution. Longer-range interactions tend to be disrupted while shorter-range interactions tend to be formed upon genome release. *p*-Values are calculated using the Wilcoxon rank-sum test. Minimum, median, and maximum are indicated as horizontal bars. **e**, **f**, **h** Top: mutations (red) are performed along a strand of the pair-wise interaction that is found dominantly in virions, as compared to in cells. Bottom: qPCR analysis of the amount of mutant and WT virus that are produced in Huh7 cells and supernatant, 24 and 48 h post infection. The Y-axis shows that amount of mutant virus produced as compared to wild type. *p*-Value < 0.05 (Student *T*-test, two-tailed). Error bars represent SEM, $N \geq 2$ for each experiment. **g** Top: mutations (red) and compensatory mutations (blue) are performed along a pair-wise interaction that is found to be abundant in both virions and cells. Bottom: qPCR analysis of the amount of mutant, compensatory mutant, and WT virus that are produced in the supernatant of Huh7 cells 24 h post infection. The Y-axis shows that amount of mutant virus produced as compared to wild type. Error bars represent SEM, $N = 6$. **i**, A129 mice ($N = 7$) were infected with $2 \times 10^5$ PFU of wild-type and mutant virus and bled at day 2, 4, 6 post infection. The viral titer in the sera was determined by plaque assay. Statistical test was performed using Mann-Whitney *U* test, ***$p < 0.001$. Error bars represent SEM, dotted line denotes the limit of detection

---

blood 4 days post infection, showing that the mutations do indeed reduces the fitness of the virus in vivo (Fig. 5i). These results confirm that the newly identified pair-wise interactions indeed play important roles during the life cycle of the viruses.

## Discussion

The DENV and ZIKV genomes are complex and play important roles in driving their pathogenesis. Here we integrated high-throughput secondary structure mapping, pair-wise interaction mapping, and evolutionary conservation information for four DENV and four ZIKV viruses to study conserved RNA structures inside virus particles and inside their host cells[24,29]. We identified: (i) many structurally and evolutionarily conserved structural elements along genomes, across different serotypes; (ii) pair-wise RNA interactions in virions and in cells, many of which can take on alternative conformations showing a large diversity of structural heterogeneity in these viruses; and (iii) dynamic and shared pair-wise RNA interactions across two or more viruses in virion particles and in infected host cells. Through in vitro cell culture and in vivo mouse experiments, we showed that some of these interactions are important for virus fitness, demonstrating the functional importance of the spatial organization over simple presence of sequence elements.

Similarly to several other viruses[6–8], structure probing of DENV and ZIKV using MaP showed that their genomes are also highly structured. While previous studies have shown an abundance of regulatory elements in the coding region of the HIV-1 genome, concentrated primarily in structured regions located between segments coding for individual proteins[6], we did not observe such a correlation for DENV or ZIKV viruses, suggesting that they likely utilize different protein processing strategies. By combining structural and sequence conservation, we were able to identify 16 and 12 structured regions in DENV and ZIKV, respectively that are potentially important; although their functional significance needs to be further verified experimentally. We note that as NAI-MaP data are generated using Illumina short read sequencing, we are currently unable to distinguish between structures derived from the shorter RNA elements (eg. sfRNA in the 3′ UTR[30]) versus structures that result from the full-length genome in cells. Further studies utilizing longer read sequencing technologies will be needed to dissect contributions of structures from shorter viral RNAs versus full-length genomes.

Using SPLASH, we observed that the majority of our pair-wise RNA interactions inside cells are shorter than 500 bases, consistent with other ZIKV structure probing data sets[28,31].

In contrast, the majority of our in virion pair-wise interactions are longer than 500 bases. This observation suggests that virus structures are either actively being disrupted inside cells[26] and/or are necessary for packaging inside the space constrained nucleocapsid. Interestingly, releasing the genome from the virus particle and performing structure probing in solution still revealed a significantly more structured genome than inside cells, suggesting that the genomes are being actively unwound inside cells. This agrees with prior literature showing that the RNA structures are less paired intracellularly, with longer interactions being preferentially disrupted[27]. Further experiments inhibiting the ribosome and the virus helicase will be needed to understand the underlying mechanisms of this observation.

In addition to finding pair-wise interactions, we also identified a large amount of structural heterogeneity in our data, consistent with previous observations of structural heterogeneity in HCV virus, and with the importance of dynamic structures in DENV and ZIKV viruses[7,28,31]. This structural heterogeneity could either correspond to variability of the genome over time, or the presence of distinct organizational patterns that occur in parallel without interconverting (such as genomes at different steps of the virus life cycle). It has previously been shown that circularized and linear forms exist in a dynamic equilibrium that is crucial for proper replication[19,32]. This may apply to these newly identified interactions as well. Further studies on probing virus structures at different time points post infection will be needed to understand the temporal dynamics of virus structures as they transition through various stages of the virus life cycle.

By using NAI-MaP reactivities as initial constraints in structure modeling and selecting structures based on subsequent concordance to our SPLASH interactions, we provide a strategy to integrate information from the two different structure probing strategies to better model RNA structures. Previous studies have used SHAPE[6,8], and/or pair-wise interaction mapping[28,31], but did not integrate both strategies into modeling. We observed that NAI-MaP constrained, SPLASH-selected structures improved modeling of 16S rRNA, indicating that it could be a general strategy to incorporate different structure probing information together.

In summary, our data expand upon the known structures in DENV and ZIKV and reveal a network of pair-wise interactions within the DENV and ZIKV genomes that likely work together to support and facilitate virus fitness. We posit that our work serves as a comprehensive resource of DENV and ZIKV genome organization, deepening our understanding of higher-order genome organization in flaviviruses. A map of the secondary RNA

structures could guide the design of broad based RNA therapeutics that target the critical RNA structures of these pathogenic viruses.

## Methods

**NAI-MaP structure probing inside virions.** DENV virus serotypes 1–4, belonging to EDEN viruses 2402, 3295, 863, and 2270 respectively (Genbank ID: EU081230.1, EU081177.1, EU081190.1, and GQ398256.1), were amplified in Vero cells and harvested from the cellular media 4–5 days post infection. ZIKV virus African strain (AY632535_MR766), French Polynesia strain (KJ776791_HPF2013), Brazil strain (KU365780_BeH815744), and the Singapore strain (KX813683_ZKA-16-097) were amplified in C6/36 cells and harvested 3–5 days post infection.

Freshly collected viruses were centrifuged at $16,000 \times g$ for 10 min at 4 °C to remove cellular debris. The viruses were then separated into three reactions: (1) we added 1:20 the volume of 1M NAI (03-310, Merck, 25 µl of NAI in 500 µl of virus in cell media) and incubated the reaction for 15 min at 37 °C for structure probing; (2) we added 1:20 volume of dimethyl sulfoxide (DMSO) to the virus and incubated the reaction for 15 min at 37 °C as negative control; and (3) we set aside a third portion of the virus as for the denaturing control in the downstream library preparation process. We then extracted all the viruses using TRIzol LS reagent (Thermo Fisher Scientific) following the manufacturer's instructions.

We performed gel electrophoresis of the virus RNA after each extraction on a 0.6% agarose gel, using ssRNA (NEB) as the ladder, to ensure that the virus RNA was intact. We typically observed a single band above the 9 kb RNA ladder that indicated the presence of intact full-length DENV and ZIKV RNA genomes. We then performed library preparation following the SHAPE-MaP protocol to generate cDNA libraries compatible for Illumina sequencing.

**NAI-MaP structure probing of Hela cells.** Hela cells (obtained from neighboring labs in GIS) were grown in Dulbecco's modified Eagle's medium high-glucose media (Thermo Fisher Scientific), supplemented with 10% fetal bovine serum (FBS), 1% Pen-Step to 70–80% confluency. Cells from a single 10 cm plate was trypsinized, washed once with phosphate-buffered saline (PBS), and resuspended in 475 µl of PBS. 25 µl of 1 M NAI were added to the cells at 37 °C for 15 min. Total cellular RNA was extracted using TRIzol extraction, followed by passing through the RNeasy column (Qiagen). We performed structure libraries for sequencing following the SHAPE-MaP protocol.

**Pair-wise interactome mapping inside virions and in solution.** For in virion SPLASH, freshly collected virus was treated with a 1:500 volume of 20 mM biotinylated psoralen (2 µl of biotinylated psoralen in 1 ml of virus in cellular media) and incubated at 37 °C for 10 min. For SPLASH on released genomes, freshly collected DENV-1 virus was treated with proteinase K (1 mg/ml), sodium dodecyl sulfate (SDS, 1%) in the presence of 10 mM $MgCl_2$ and 100 mM NaCl, at 37 °C for 30 min. Biotinylated psoralen (200 µM) was then added to the treated virus particles at 37 °C for 10 min. The treated viruses were then spread onto a 10 cm dish and ultraviolet (UV)-irradiated at 365 nm for 20 min on ice to crosslink the interacting regions. The crosslinked virus genomes were then extracted using TRIzol LS reagent (Thermo Fisher Scientific) following the manufacturer's instructions. We performed SPLASH libraries similarly to the published protocol in Aw et al.[24,33].

**Interactome mapping of viruses inside human cells.** Huh7 and human neuronal precursor cells (obtained from neighboring labs in the Genome Institute of Singapore and Duke NUS medical school) were infected with DENV1 and ZIKV viruses at an multiplicity of infection (MOI) = 1 for 30 and 24 h respectively. The cells were washed twice with PBS. The cells were then incubated with 200 µM biotinylated psoralen and 0.01% digitonin in PBS for 10 min at 37 °C, and irradiated at 365 nm of UV on ice for 20 min[33]. Total RNA was then extracted from the cells using TRIzol reagent according to the manufacturer's instructions. We performed SPLASH libraries similarly to the published protocol in Aw et al.[24,33].

**Functional assay to test for virus fitness.** The entire genomes of DENV-1 and ZIKV French Polynesia viruses were cloned into five overlapping fragments. Mutagenesis was performed on individual fragments and the genome was then assembled using Gibson assembly (NEBuilder HiFi DNA Assembly Master Mix) by mixing 0.1 pmol of the five fragments with a vector fragment together. The Gibson assembled fragments were then transfected into Human Embryonic Kidney 293T (HEK293T) cells (obtained from neighboring labs in GIS) using Lipofectamine 2000 (Thermo Fisher Scientific). We collected the media from the HEK293T cells after 72 h and infected C6/36 cells (obtained from E.E.O.'s lab in Duke NUS) to amplify the virus. We collected the supernatant from C6/36 cells after 5 days post infection and performed plaque assay to determine the virus titers. We then normalized all the wild-type and mutant viruses and infected Huh7 cells at an MOI = 1 and MOI = 0.1. Both the supernatant and the cellular RNAs were harvested after 24 and 48 h and the amount of virus inside the cells and released into the supernatant was determined by quantitative PCR. To confirm that the mutant virus sequence was indeed what we engineered it to be, we performed reverse transcription-PCR on DENV and ZIKV genomes, using RNA extracted from infected Huh7 cells at 48 h post infection. We then performed Sanger sequencing of the full virus genome to confirm the sequence of the mutant.

**Mouse experiments.** Five-week-old A129 mice (129/Sv mice deficient in the alpha/beta (interferon-α/β) interferon receptors; $n = 7$) were infected via the subcutaneous (s.c.) route with $2 \times 10^5$ plaque forming units of ZIKA H/PF/2013 WT and mutant. Mice were bled at day 2, 4, and 6 post infection and the virus titer in the blood serum was determined by plaque assay.

For the plaque assay, baby hamster kidney-21 (BHK-21) cells were seeded into 24-well plates. The collected supernatants were serially diluted fivefold in RPMI 1640, 2% FBS (starting from 1/10), and added to the seeded cells. After an hour incubation at 37 °C, 1% (w/v) carboxymethyl cellulose (containing RPMI media and 2% FBS) was added to the wells. After incubation at 37 °C for 4 days, the cells were fixed with 4% paraformaldehyde and stained with 1% crystal violet for a minimum of 1 h. The plates were then thoroughly rinsed with water, before plaques were visually enumerated.

All the animal experiments were carried out under the guidelines of the National Advisory Committee for Laboratory Animal Research in the AAALAC-accredited NUS animal facilities. The animal experiments described in this work were approved under the NUS Institutional Animal Care and Use Committee protocol number 16-00263. Non-terminal procedures were performed under anesthesia, and all efforts were made to minimize suffering.

**Computational analysis of NAI-MaP mutation rates.** After quality filtering with a Phred cutoff of 25, reads were aligned against the published reference genomes for DENV serotypes 1–4 and the four ZIKV strains using Bowtie 2 at the highest default sensitivity setting[34]. These alignments were performed separately for the data sets from experiments treated with DMSO and NAI respectively, as well as the denatured experiment. Mutations were counted separately in each set and subsequently expressed as a shape score by calculating $(M_{NAI} - M_{DMSO})/M_{denat}$ at each position as outlined by Weeks and co-workers[17]. We calculated the distribution of continuous segment lengths and segment lengths with two transitions as would be expected for hairpin or helices containing a bulge. We compared these metrics with a variety of other types of RNA such as Xist mRNA, 28S rRNA, and random controls[35]. Significance levels in the difference of distributions was calculated using the Wilcoxon rank-sum test.

**Sequence analysis of different DENV and ZIKV genomes.** We performed multiple sequence alignments (MSAs) of the reference genomes using the program MAFFT with the –genafpair option[36] to align the various virus strains and species to one another. The reference alignment is provided as supplementary material. Subsequently, a MSA score was calculated for each position by determining highest number of identical pairs among all sequences and dividing it by the number of possible pairs at that position, i.e. if 7 out of 8 sequences were identical at position 1234, then the MSA score would be calculated as 21 (number of identical pairs in the 8 sequences)/28(number of unique pairs in 8 sequences) = 0.75. We then analyzed the degree of sequence conservation at locations classified as high-reactivity (>0.5) vs. low-reactivity (<0.5) according to NAI-MaP reactivity.

**Analysis of covariation within viral species.** We retrieved all full genome sequences for DENV1–4 and ZIKV from the Virus Pathogen Resource (https://www.viprbrc.org), resulting in 1850 (DENV1), 1420 (DENV2), 923 (DENV3), 220 (DENV4), and 540 (ZIKV) full genome sequences respectively. After aligning the sequences for each viral species separately using mafft with the -localpair option, we created a merged alignment using the mapping feature of mafft. This alignment is provided as a supplementary data file. We proceeded to use this alignment as an input for R-scape (https://doi.org/10.1038/nmeth.4066) with a window size of 400 nucleotides (nt) in 100-nt increments, and used the GTp statistic to retrieve covariations at positions proposed by our structural models. It should be pointed out that for DENV4 there was only a small number of full genome sequences available and so was somewhat underrepresented in the data set. For ZIKV, the set of full genome sequences contains a high number of, and similar sequences, mostly originating in the recent South American outbreak, which exhibit narrow variability and hence may bias the ZIKV portion of the data to a specific subtype not necessarily representative of all ZIKV viruses. Across the manuscript, evidence of covariation is indicated by green circles for the respective base pairs.

**Structure and sequence conservation across virus genomes.** To assess the relationship between structure and sequence conservation, structure similarity was compared with sequence identity in 100-nt segments with a step size of 10 nt (Fig. 1d). As expected, sequence and structure similarity is correlated. Interestingly, there exists a set of regions that exhibit higher structure conservation than would be expected based on sequence conservation alone. (Fig. 1e, red). We localized these regions within each genome and found that among all different pairs of DENV serotypes the higher-than-expected structure conservation regions cluster in specific areas (Fig. 1d, bottom). To assess whether structures are conserved across DENV and ZIKV viruses, reactivities were classified into high (>0.5, H) and low (≤0.5, L) and majority votes were applied at each position of the reference alignment. Bases in the four DENV subtypes or the four ZIKV strains that have either 3/4 or 4/4 identical shape classes (e.g. HHHL would be classed as H, LLHL would be

classed as L) were as classified as structurally similar. Classes without a majority were identified as indeterminate (e.g. HLHL or HHLL) and were not included in the subsequent analysis. Subsequently, we divided the genome into positions that have identical majorities for DENV and ZIKV (same, S), and positions where the majority structure is different between DENV and ZIKV (different, D). We then calculated the distribution of MSA scores within each class (Supplementary Figure 3c,d). Significance levels in the difference of distributions was again assessed using the Wilcoxon rank-sum test.

**Nominating functional structures along virus genomes**. Regions of functional interest were identified if they fulfilled at least two out of three of the following criteria: (i) structure similarity is in the top 30% among all regions; (ii) structured stretch length is in the top 30% of all regions; and (iii) the region is in the bottom 30% for synonymous mutation rates, indicating selection pressure on the RNA itself for both DENV and ZIKV. Structural similarity was assessed by calculating the local identity of the discretized shape scores. The length of structured stretches was calculated from the same data set. The synonymous mutation rate was calculated from an alignment of all DENV and ZIKV full genome sequences available at the time of writing. Structure models for all identified sequences including available covariation data are presented in the manuscript.

**Structure modeling of DENV and ZIKV genomes**. Using the Shape-MaP data in conjunction with RNA structure prediction algorithms allowed us to create structural models for the full-length DENV and ZIKV genomes. We utilized the program RNAstructure[37] and incorporated the Shape-MaP results as additional restraints using a slope of 1.1 and an intercept of 0.0 kcal/mol for the Shape-MaP potentials. These values were obtained by optimization of structure prediction for the respective known 5′ and 3′ structural elements of each virus using NAI-MaP data[10,18,19,38], otherwise using default settings. Partition functions were calculated using *partition-smp* from the RNAstructure software package and were subsequently used to calculate pairing probabilities. Pairing probabilities were used to calculate Shannon entropy as a metric of prediction confidence (Fig. 1b) at each base position. Subsequently, the partition function files were processed using *stochastic-smp* to obtain an ensemble of 1000 structures. From the ensemble, we applied different selection criteria to obtain structures that fit with specific experimental data obtained in this study, and presented the most congruent structures. The structural ensembles for all viruses are provided as supplementary data. We clustered the structures of the ensembles by calculating pair-wise similarity and performing principal component analysis. The resulting distributions group similar structures close together. We then indicated the structures best predicting the known 5′/3′ structural elements, as well as structures agreeing best with long-range interaction data in these plots (Fig. 2b, Supplementary Figure 7), and present the corresponding structures in arc plot representation to visualize the structural diversity encountered in our data.

**Analyzing pair-wise interactions identified by SPLASH**. SPLASH reads were aligned to their respective reference genome using the program Bowtie 2 to perform a paired read alignment[34]. A matrix containing a count of all pair-wise T positions observed in the reads was compiled. To filter random pairs, we constructed a baseline matrix by randomly shuffling all read pairs 100 times and subsequently subtracted this baseline matrix from the raw read matrix. We then considered all sites that exceed a threshold value of 10% of the maximal observed value as valid SPLASH interactions. The shuffling procedure and threshold values were optimized for the known human and *Escherichia coli* ribosome structures, and allowed us to markedly improve specificity and the positive predictive value for SPLASH interactions. Long-range interactions were considered conserved if they occured in at least two strains. Plotting the precise read depth across the genome allows for identification of distinct peaks mostly within <50-nt segments and thus allows the localization of the interaction sites. We analyzed the promiscuity of these long-range interactions by counting the number of alternate interaction partners we found at each site originating long-range interactions. We identified several sites that show multiple coinciding interactions for the same sequence stretch. Competing interactions were modeled using *RNAcofold* from the ViennaRNA package and visualized including the Shape-MaP and covariation data[25]. We then proceeded to analyze the structural ensembles obtained from NAI-Map constrained modeling to select the structures most congruent with SPLASH data. To obtain alternative structures, we proceeded to subtract all SPLASH interactions associated with the best match out of the data set and repeated the procedure to obtain the second best match with SPLASH data. These structures are presented in the Supplementary Figure 8.

**Peak calling of SPLASH interaction sites**. The genome of each of the eight viruses were divided into non-overlapping bins of 100 bases. Each chimeric read was split and mapped to two paired bins. Paired bins that contain at least 20 chimeric reads were then selected for peak calling. The density curves of read coverage were fit (using the density function of R) for the left and right bins respectively. The peak of the interaction was determined to be the position with the maximum coverage density.

**Analysis of closest interaction pairs in different viruses**. Every pair-wise interaction was analyzed as a point in two-dimensional (2D) space (i.e. the 1200–7000 interaction would be assumed to have the coordinates [1200,7000]). To determine if there were other similar interactions nearby in other virus serotypes, we identified the closest point in a different virus 2D map (e.g. the nearest long-range interaction of DENV2 for a specific interaction in DENV1) to a specific serotype. We then compared the distribution of these closest distances with distributions calculated from random 2D maps with the same number and distribution of points as our SPLASH interactions, and calculated the significance levels between these sets. Highly similar viruses such as ZIKV-SIN, ZIKV-FP, and ZIKV-BR showed that the average distances to a related interaction are very short, indicating a high level of conservation. The *p*-value was calculated using the Wilcoxon rank-sum test.

**Reporting Summary**. Further information on experimental design is available in the Nature Research Reporting Summary linked to this article.

## Data availability

The data sets generated during and/or analyzed during the current study are available in the GEO repository, GSE106483. The authors declare that all other data supporting the findings of this study are available within the article and its Supplementary Information files, or are available from the authors upon request.

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

## Acknowledgements

We thank members of the Wan lab, Sessions lab, Ooi lab, Meng How Tan, Mariano Garcia-Blanco, and Ben Blencowe for discussions. Y.W. is supported by funding from A*STAR and Society in Science—Branco Weiss Fellowship. P.J.B. acknowledges support by funding from the Ministry of Education Singapore—MOE AcRF Tier 3 Grant Number MOE2012-T3-1-008. R.G.H. acknowledges support by funding from the A*STAR Biomedical Research Council—Young Investigator Grant BMRC YIG 1610651031. E.E.O. is supported by funding from the National Medical Research Council (NMRC) Senior Clinician-Scientist Award (NMRC/CSA/0060/2014) and Individual Research Grant (NMRC/CIRG/1462/2016).

## Author contributions

Y.W. conceived the project. Y.W., E.E.O., O.M.S., P.F.S., S.A. and X.J.L. designed the experiments. X.N.L., J.G.A. and S.Y.L. performed the NAI-MaP and SPLASH experiments. W.C.N., M.M.C., H.X.P., T.C., P.F.S. and K.B.S. performed the virus experiments. A.Y.L.S., R.G.H., Y.S., A.W. and P.J.B. performed the computational analysis. X.S. generated the human neuronal precursor cells. H.K.T. and P.H.B. performed the mouse experiments. L.J. performed chemical synthesis to make NAI. Y.W. organized and wrote the paper with A.Y.L.S., R.G.H., O.M.S., P.F.S., M.H. and N.N. and all other authors.

## Additional information

**Competing interests:** The authors declare no competing interests.

