## [Peer Review File · Nature Communications]

Reviewers' Comments:

Reviewer #3:

Remarks to the Author:

Referee's report: "Structure mapping of dengue and Zika viruses reveals new functional long-range interactions" by Huber et al.

Zika and dengue viruses are representative of a large group of mosquito-borne flaviviruses that are ongoing health threats. As with all single-stranded positive-sense RNA viruses, the viral genomic RNA contains cis-acting structured elements that are important for controlling various processes during viral infection. In addition, long-range interactions within these RNA genomes are important. This mandates studies to identify and characterize both local structured elements and long-range interactions, discover their function, and describe their mechanism.

In this manuscript, Huber et al all present chemical probing and crosslinking data on the entire viral genomes of the Zika and dengue viruses, both in virions and in cells. They use these data and some sequence comparisons to propose the presence of several local structural elements and long-range interactions. They make mutations designed to disrupt some of these structures and interactions and observe changes consistent with these structures being important to the virus. They also assess the effect of translation on the structure of the viral RNA.

Overall, the chemical probing data are of high quality. Many of the putative structures and interactions that they observe are compelling, and if properly presented and interpreted, I think these data have much to offer the field. However, my enthusiasm is dampened by the fact that the paper is a bit of a "data dump" that hits on several topics, each fairly superficially. There is a lot of data, in fact I would argue too much for a single paper. As a result, the main points are left underdeveloped and not fully supported, controls are missing, and controls that are present are difficult to assess. While it is not my place to tell the authors how and what to publish, my advice would be to strongly consider using these data in 2-3 papers, each fleshing out a key discovery; that would raise the impact of the discoveries because right now the analyses of each point are fairly superficial.

Specific Points:

1. The authors are quite confident about the "correctness" of their structure, and to a certain degree they are building on this term being applied in papers by others in the field. But this is dangerous because it gives the false impression that we now know what THE structure of this RNA is. In fact, there certainly are many different structures co-existing. The authors do mention that they see alternate structures for 40% of the RNA, this fact needs to be discussed and the data and resultant structures interpreted in light of this observation. I do not understand how the variation fits with the single 2D model shown in Figure 2. When they are talking about alternate structures, do they mean alternate long-distance interactions but conserved local 2D structures? Or is variation found in local region but not in long-range interactions? Or are they coupled? Are there some structures that are mutually exclusive between various alternate conformations? In the 2D model of Figure 2, how should I think about chemical probing that does NOT fit with the 2D model (easily spotted)? Is this indicative of alternate structures? This is all unclear because the interpretation and discussion is very shallow.

2. Moving from the global model to specific interactions or structured elements, there is again a need for more detailed analysis and discussion. At a minimum, for each new element identified, it is important to show whether the newly proposed structure is supported by sequence alignments. There is a rich repository of sequence data from viral isolates of all kinds of flaviviruses which could readily be used. (hundreds from dengue alone). To be fair, this is shown to some extent on Fig 2b, but there is it mostly the 3'UTR that's convincing (which we already know a lot about). I am missing the alignments needed for me to really assess the validity of the reported covarying base pairs. Apparent covariation can occur in random sequences; thus, many sequences are needed to assess this, and that analysis should be present somewhere for me to judge. Conversely, if these regions of the RNA are too highly conserved for much covariation, then the reader needs to be able to see how conserved. It looks like they have access to all that information (extended data 3 and 4).

3. There is a real need to verify that the mutations made have the structural effect that is predicted. When making mutations to alter RNA structure, it is the accepted practice in the field to first assess that the mutations have the predicted or intended effect by repeating the structural or biophysical analysis (in this case, probing) with the mutant. If one "side" of a long-range interaction is altered, then probing should show a change in the pattern there and in the other "side" of the interaction. This validates the interaction and the mutation and allows the functional effects of the mutation to be properly interpreted. These are essential controls.

4. Another validation tool could be to calculate correlation coefficients between the predicted structures and those that we already know of based on prior studies (especially for the 3' UTR). That would give a confidence criteria for sequences elsewhere on the genome for which we had no prior reference. Related, I am concerned that the authors present the secondary structure of the 3' UTR for reference, but parts of this 2D structure are not correct based on published, validated structures (including 3-D crystal structures!). If this is not correct, what else is not correct?

5. The data dealing with unfolding of the RNA genome in the cell, and the role of the ribosome and the action of helicases is underdeveloped. To a certain degree, this is obvious – we know that the viral genome is the template for infection processes in the cell that must unwind it (replication and translation) and we know that it exists at some point as a fully double-stranded intermediate; hence the conclusion that "the structured state we observe in side cells is a result of active unwinding by enzymes..." and that the viral replicase machinery could play a roles not novel is a bit odd because we knew this...its settled virology. So, I'm left wondering: what did we learn here?

6. Related to the above, infection by these RNA viruses is a very dynamic process in which the RNA plays several roles. In addition, these roles change over time and thus the amount of RNA in a given population (translating, replicating, packaging) also changes. In addition, these viruses generate large amount of several non-coding subgenomic RNAs that contain part of the genomic RNA sequence (I do not think this is even mentioned but it is a critical point!), hence there are at least 2-3 different RNA species that share sequence during infection. Each may have different structures and be doing different things. The authors do mention different structural states existing, but this is not fleshed out and thus I am left feeling like I don't really know much more about the reality of the structure of these RNAs in the cell than I did before. While the authors do not need to have all the answers, they need to talk about these data in light of these virological realities, otherwise it is superficial. This could be a paper in and of itself, if properly controlled and discussed.

7. Overall, the discussion section leaves many mysterious and unanswered questions. I'll just say that two paragraphs is not nearly enough to cover all the caveats, various interpretations, integration with known virology and RNA structure, etc. given the density and amount of data presented.

Minor points

Other points:

1. Fig 1c+d: I think clustering what's <0.5 separately from what's >0.5 hides away information that would be best covered by a spectrum of three or more colors. I don't think that a reactivity of 0.49 for example would correspond to a paired residue, although here it would be black. And how many reactivities are in that 0.3-0.7 range anyway? Better to stick to the same scale as on Fig 2b.

2. lines 125-127: "As expected, regions that share similar structure patterns between the 8 viruses generally share higher sequence identity, consistent with the idea that sequence plays an important role in RNA structure". It's pretty obvious that sequence is important for structure formation so this statement comes off as odd.

3. line 146: Converting SHAPE reactivity to pseudo-energy terms has been shown by others to not be appropriate, unless that pseudo energy term is not regarded as an energy, but as a probabilistic inference. It's mostly useful in the context of such long RNAs when shown to support structures derived from other methods, such as sequence alignment, or probing using other probes (see Eddy SR, *Annu Rev Biophys* 43:433-456 (2014)).

4. Fig 2b: I see many regions with problems in there: see 69-88, 782-802, 2247-2275, 4090-4120, either they are paired and should not be shown as unpaired, or vice-versa. What's going on here?

Reviewer #4:

Remarks to the Author:

The Authors have adequately addressed the concerns of Reviewer-1. Overall, the reworked manuscript provides several novel insights into the global genome organization of flaviviruses. Importantly, candidate long-range RNA-RNA interactions were tested for functionality, and were shown to be relevant. The study is well suited for the audience of this journal.

There are, however, a few issues that require attention prior to further consideration.

Line 47 – Abstract - change "the importance of these new" to "the importance of one of these new", because only one interaction was assessed by compensatory mutational analysis

Line 78 – change "essential" to "important", because the mutant viruses were still able to replicate to readily detectable levels

Fig2b – explain in legend what is signified by the lighter shades of red, orange, and black in the color key

Line 236 – "exact.....sequence identities across serotypes" – in Fig4d (upper and middle panels) green circles indicate covariation and thus substitutions within the interaction in different serotypes – therefore, "exact" sequence identities were not maintained for these two examples – thus, this statement needs correcting

Fig5f,g,h,I – needs statistical evaluation – there are many very large error bars

Fig5h has the results for the compensatory mutant (gold bars) in the graph, but it is MISSING results from the "mutant" (blue bars) containing substitutions in only the lower sequence in Fig5h - a critical piece of information.

Reviewer #5:

Remarks to the Author:

I was asked to comment on the rebuttal to ref#2. I have the following comments:

General comments: I agree with ref#2 that the paper suffers from lack of functional studies. I do not think the authors provide sufficient new experiments to fulfil this shortcoming. The topic of long-range RNA interactions in viral genomes is interesting and worth studying. The RNA structure mapping in this paper is technically advanced and large amounts of relevant data are generated. The description is largely technical, sometimes difficult to follow, and I find the nice-looking Figures not easy to interpret. In my opinion where the paper falls short is, the interpretation of the results and subsequent experimentation focused on structure-function (very basic characterization of virus mutants). These limited functional studies are not very insightful. Recoded viral genomes may not only change RNA structures, but will also impact codon usage, codon bias, dinucleotide bias, all of which may affect replication in their own right. Vice versa, it is also possible to scramble the RNA of entire viral RNA genomes (while keeping codon usage and CpG bias the same) without major effects on replication – thus one could argue that the RNA structures in between the conserved UTRs are biologically not so important. For an example, see Fros et al., 2017 eLife. Without a better in-depth understanding of the biological importance of the ZIKV and DENV RNA structures, the paper is largely descriptive and it is not yet clear what these data mean for our understanding of the function of these RNA structures.

It would have been beneficial if the authors, in addition to ZIKV and DENV, would have performed a control with a flavivirus that has known internal RNA structure, e.g. WNV with a ribosomal frameshift in the end of NS1, leading to NS1' (see Firth papers for these structures).

Comment 1 and 2. I did not have sufficient experience with ribosome profiling to comment.

Comment 3. The structures have been shown as requested. The authors have also replaced the mutagenesis experiments with new experiments in which only point mutations disrupting RNA structures were tested. These viruses have been sequence verified. This is a better approach, but still does not provide the level of understanding that would be needed to propose that these RNA structures are indeed important and why. It is not entirely clear why some viruses are attenuated in vertebrate cells but not in invertebrate cells and vice versa. This at least would suggest that other mechanisms than RNA structure alone account for this difference, e.g. dinucleotide bias, which affects viruses in vertebrate but not invertebrate cells.

Comment 4. The authors responded satisfactorily to this comment.

We thank the reviewers for their helpful comments. We have now performed additional experiments and analysis to strengthen our manuscript, including demonstrating that one of our virus mutants is attenuated in vivo in a mouse model. To integrate NAI-MaP reactivities with pair-wise RNA interaction information to model RNA structure ensembles, we now constrain structures based on NAI-MaP reactivities and select for structures with the largest concordance with SPLASH interactions. We tested that combining these two strategies indeed improves structure modelling in known RNA structures such as 16S rRNA. Furthermore we also show that our high throughput interactome structure data has high concordance with high throughput interactome datasets from two other independent groups, indicating that we are robustly and reproducibly able to detect new pair-wise RNA interactions in dengue and Zika genomes. We additionally show that: 1) our NAI-MaP structure mapping data is of good quality, by mapping to known structures in the 5' and 3'UTRs; 2) NAI-MaP constrained, SPLASH selected structures are enriched for co-varied bases; 3) both long and short pair-wise interactions have similar propensity to form alternative structures, although shorter range interactions are more conserved across viruses and inside cells; and 4) that our mutations indeed disrupt the interactions as they should, through pulldown and qPCR experiments. With these deeper analyses and controls in place, we have substantially revised our manuscript to show that our data is technically sound and that the long-range interactions we have identified are biologically relevant, enabling this manuscript to be a useful resource for the community.

Reviewers' comments:

Reviewer #3 (Remarks to the Author):

Referee's report: "Structure mapping of dengue and Zika viruses reveals new functional long-range interactions" by Huber et al.

Zika and dengue viruses are representative of a large group of mosquito-borne flaviviruses that are ongoing health threats. As with all single-stranded positive-sense RNA viruses, the viral genomic RNA contains cis-acting structured elements that are important for controlling various processes during viral infection. In addition, long-range interactions within these RNA genomes are important. This mandates studies to identify and characterize both local structured elements and long-range interactions, discover their function, and describe their mechanism.

In this manuscript, Huber et al all present chemical probing and crosslinking data on the entire viral genomes of the Zika and dengue viruses, both in virions and in cells. They use these data and some sequence comparisons to propose the presence of several local structural elements and long-range interactions. They make mutations designed to disrupt some of these structures and interactions and observe changes consistent with these structures being important to the virus. They also assess the effect of translation on the structure of the viral RNA.

Overall, the chemical probing data are of high quality. Many of the putative structures and interactions that they observe are compelling, and if properly presented and interpreted, I think these data have much to offer the field. However, my enthusiasm is dampened by the fact that the paper is a bit of a "data dump" that hits on several topics, each fairly superficially. There is a lot of data, in fact I would argue too much for a single paper. As a result, the main points are left

underdeveloped and not fully supported, controls are missing, and controls that are present are difficult to assess. While it is not my place to tell the authors how and what to publish, my advice would be to strongly consider using these data in 2-3 papers, each fleshing out a key discovery; that would raise the impact of the discoveries because right now the analyses of each point are fairly superficial.

We thank the reviewers for his/her comments. We have now deepened our analysis and done more controls to show that our experiments are technically sound and have removed the ribosome profiling experiment to streamline the paper. We detail our responses below:

Specific Points:

1. The authors are quite confident about the “correctness” of their structure, and to a certain degree they are building on this term being applied in papers by others in the field. But this is dangerous because it gives the false impression that we now know what THE structure of this RNA is. In fact, there certainly are many different structures co-existing. The authors do mention that they see alternate structures for 40% of the RNA, this fact needs to be discussed and the data and resultant structures interpreted in light of this observation. I do not understand how the variation fits with the single 2D model shown in Figure 2.

We thank the reviewer for these insightful comments. We have now integrated both local secondary structure information and pair-wise RNA interactome information into structure modeling by using NAI-MaP reactivities as structural constraints to generate an initial ensemble of 1000 potential structures. We then selected the top 10 structures based on the best fit with our SPLASH interactome data. We confirmed that NAI-MaP constrained and SPLASH selected structures are significantly enriched for correct structures, based on 16S and 18S rRNA structures ($p < 0.003$). Top NAI-MaP constrained, SPLASH selected structures are present in the majority of the structural clusters, representing the diversity of structures present in dengue

Figure 1. Left: Clustering of 1000 structure ensembles based on the NAI-MaP constraints. The green dot represents the structure with the most accurate 5' and 3' UTRs, the red and yellow dots represents the structure with the largest and second largest concordance with SPLASH interactions respectively. The black dots represent the structures with the top 10 best concordance with SPLASH. Right: Arc plots showing the pair-wise interactions in each structure selected based on NAI-MaP constraints and either best 5'/3' UTR correctness, largest or second largest concordance with SPLASH.

and Zika. We have now added this data as Extended data 8 in the manuscript.

For the sake of clarity in representing how dengue/Zika structures may look like inside virion particles, in Figure 2b in the manuscript we have presented the structure models for those with the best fit to known 3' and 5' UTRs in dengue and Zika. We have now clarified in the manuscript that this is just one potential structure model out of an ensemble. We do note that our 16 structurally conserved regions in DENV-1 are enriched for low Shannon entropies (Extended data 4d), suggesting that these elements are more likely to show unique structures within the ensembles, and hence may represent more stable structures.

Figure 2. Modeled DENV-1 structures along the genome using NAI-MaP as constraints in structure modeling. The grey boxes indicate the 16 potentially structurally important elements in the dengue genome (Figure 2a,b).

When they are talking about alternate structures, do they mean alternate long-distance interactions but conserved local 2D structures? Or is variation found in local region but not in long-range interactions? Or are they coupled? Are there some structures that are mutually exclusive between various alternate conformations?

We thank the reviewer for these comments. When we talk about alternate structures, we do not distinguish between short (<500 bases) or long (>=500 bases) pair-wise interactions. To test whether shorter- or longer-range interactions tend to be more variable and hence form more alternative structures, we calculated the proportion of short vs long interactions that are involved in alternative base pairing. We observed that both shorter and longer range interactions tend to form similar numbers of alternative structures, although longer interactions have a slight tendency to form more interactions (48.8% in >500bases vs 41.6% in <500 bases, $p=0.12$, Figure 3c in the manuscript). Interestingly, we noticed that in cell interactions tend to be either unique or have only one alternative interaction (7.8% of in cell interactions form more than one alternate structure and 14.8% of in virion interactions form more than one alternate structure, $p=0.02$), while in virion interactions tend to show greater heterogeneity, suggesting that the genomes exist in a greater diversity of conformations inside virions.

Figure 3. Percentage of long (≥ 500 bases) and short (< 500 bases) interactions that forms unique or alternative interactions in virions and inside cells, for both dengue and Zika viruses. Both long and short interactions have similar propensity to form alternate interactions, although in cell interactions tend to form fewer than >2 alternate pairings (7.8% in cell vs 14.8% in virion, $p=0.02$)

To test whether shorter or longer interactions tend to be more conserved, and are hence shared across two or more viruses, we calculated the fraction of long vs short interactions that are shared in >2 dengue or Zika strains. Interestingly, we observed that the shorter (<500 bases) interactions tend to be more conserved across viruses. We have now added this data as Extended data 7a in the manuscript.

Figure 4. Short range (<500 bases) pair-wise interactions have a higher propensity than long range interactions (≥ 500) to be shared across two or more viruses in virions (11.2% and 39% of long interactions in dengue and Zika are conserved respectively, while 42% and 55% of short interactions are conserved.) Zika virus has a higher proportion of interactions that are shared across viruses due to the high sequence similarity between the Zika strains, as compared to dengue.

Similarly, we also observed that shorter interactions inside virions are preferentially conserved in cells, indicating that shorter interactions are less variable.

Figure 5. Short range (<500 bases) pair-wise interactions in virions have a higher propensity than long range interactions (≥ 500) to be shared inside cells (18% and 36% of long interactions in dengue and Zika are conserved respectively, while 42% and 61% of short interactions in virions are found inside cells.)

In the 2D model of Figure 2, how should I think about chemical probing that does NOT fit with the 2D model (easily spotted)? Is this indicative of alternate structures? This is all unclear because the interpretation and discussion is very shallow.

We thank the reviewer for these comments. The NAI-MaP reactivity reflects the probability that a base is paired or unpaired, but does not strictly indicate structuredness. As such, not all of the unpaired bases will have high reactivity, such that the mode of SHAPE-reactivity for unpaired bases is still at 0¹. We plotted the distribution of our reactivity in known paired and unpaired bases in the 5' and 3'UTRs of dengue and Zika viruses and saw that known unpaired bases tend to have higher reactivity profiles, as expected from the Eddy SR 2014 paper. We have now added this data as Extended data 2f in the manuscript.

Figure 6. Top, distribution of NAI-MaP reactivities from our data sets in bases that are paired (left) and unpaired (right) in previously described 5' and 3'UTR structures of dengue and Zika. Unpaired bases have a greater fraction of bases with high NAI-MaP reactivity, although the mode of the distribution is also at 0. Bottom: Likelihood ratio of a base being paired or unpaired versus NAI-MaP reactivities. We obtained a similar relationship as previously described by Eddy¹.

To determine whether chemical reactivities that do not fit the structure model could be a result of alternate pairings, we calculated the distribution of Shannon entropies for bases that fit the model, versus the bases that do not, as bases with low Shannon entropies are indicative of unique structures. Interestingly, we observe that bases that do not fit the model tend to have higher Shannon entropies, suggesting that they may be involved in alternate pairings. We have

Figure 7. Boxplot of the distribution of Shannon entropies in bases that fit the 5' and 3'UTR structure models in dengue and Zika viruses. Bases that do not fit the known structure models tend to have higher Shannon entropies.

now added this data as Extended data 5c.

2. Moving from the global model to specific interactions or structured elements, there is again a need for more detailed analysis and discussion. At a minimum, for each new element identified, it is important to show whether the newly proposed structure is supported by sequence alignments. There is a rich repository of sequence data from viral isolates of all kinds of flaviviruses which could readily be used. (hundreds from dengue alone). To be fair, this is shown to some extent on Fig 2b, but there is it mostly the 3'UTR that's convincing (which we already know a lot about). I am missing the alignments needed for me to really assess the validity of the reported covarying base pairs. Apparent covariation can occur in random sequences; thus, thus many sequences are needed to assess this, and that analysis should be present somewhere for me to judge. Conversely, if these regions of the RNA are too highly conserved for much covariation, then the reader needs to be able to see how conserved. It looks like they have access to all that information (extended data 3 and 4).

We thank the reviewer for his/her thoughtful comments. We have now performed covariation analysis using ~5000 sequences from dengue and Zika genomes, using the program R-Scape². Our NAI-MaP constrained structure models are enriched for co-varied bases (418-450 covaried bases for NAI-MaP constrained structures vs 258 covaried bases in the shuffled structure), as compared to shuffled sequences, providing support for the modelled structures. We have now provided an example of co-varied bases in a newly identified alternative pair-wise RNA interaction (from Figure 3) and have now uploaded the alignments as a supplementary data file

Figure 8. An example of the covariation analysis for one of the newly identified pair-wise interactions in Zika French Polynesia genome. Distribution of all of the potential nucleotide pairs across ~5000 DENV and ZIKV full genome sequences are shown (the alignment is provided as a supplementary data file). ZIKV sequences account for about 500 out of 5000 sequences.

1.

3. There is a real need to verify that the mutations made have the structural effect that is predicted. When making mutations to alter RNA structure, it is the accepted practice in the field to first assess that the mutations have the predicted or intended effect by repeating the structural or biophysical analysis (in this case, probing) with the mutant. If one “side” of a long-range interaction is altered, then probing should show a change in the pattern there and in the other “side” of the interaction. This validates the interaction and the mutation and allows the functional effects of the mutation to be properly interpreted. These are essential controls.

We thank the reviewer for these comments. We have sequenced the mutants by capillary sequencing to make sure that the mutations are correct in Zika viruses. In addition, we have also performed an experiment in which, if one “side” of a long range interaction is mutated, the other “side” will not be able to interact as well. As such, if we fragment the Zika genome, pull down one side of the mutated interaction, and qPCR for the presence of the other side, we may determine whether the mutation has disrupted the interaction. Our experiments show that upon fragmentation, qPCR of one mutant strand results in the poorer pull down of the interacting strand, validating that the pair-wise RNA interactions indeed exist in the host cells and that the mutations disrupt the pair-wise interaction as expected. We have now added this data as Extended data 9c in the manuscript.

Figure 9. Testing that the mutations disrupt the RNA pair-wise interactions as expected. We performed pull-downs using biotinylated probes against one strand of the pair-wise interaction and performed qPCR analysis on the other complementary strand. Mutations on one strand of the interactions disrupt the pairing, and result in a lower amount of the complementary strand being pulled down. We tested two mutants that we validated in Figure 5 of our manuscript and showed that both mutations disrupted the pairwise interactions.

4. Another validation tool could be to calculate correlation coefficients between the predicted structures and those that we already know of based on prior studies (especially for the 3' UTR). That would give a confidence criteria for sequences elsewhere on the genome for which we had no prior reference.

We thank the reviewer for his/her comments. We have previously shown that integrating NAI-MaP reactivities into structure modeling improves the accuracy of our modelled 16S and 23S rRNA structures (Extended data 5b). We have now revised our predicted structure models by optimizing the parameters used for including SHAPE data into the structure modeling procedure and generating an initial ensemble of 1000 potential structures. The structures with the best concordance to 5'/3' UTRs show 91%, 92%, 78% and 91% accuracy (modelled dengue1-4 structures respectively) compared to the known dengue 2 5'UTR structure, and an accuracy of

76%, 81%, 78% and 85% (for modelled Zika Africa, Brazil, French Polynesia and Singapore strains, respectively) compared to the Zika Brazil 5'UTR structure. We see a concordance of 87%, 93%, 86% and 63% accuracy (modelled dengue1-4 structures respectively) to the known dengue 2 3'UTR structures, and 73%, 84%, 78%, 83% accuracy (Zika Africa, Brazil, French Polynesia and Singapore respectively), compared to the Zika Brazil 3'UTR structure. As most of the known structures are focused on dengue 2 virus and the Brazil and French Polynesia strain of Zika, lower similarities between predicted strains and known structures could also be due to differences in actual structures between the viruses. We have now included this information in the manuscript.

Related, I am concerned that the authors present the secondary structure of the 3' UTR for reference, but parts of this 2D structure are not correct based on published, validated structures (including 3-D crystal structures!). If this is not correct, what else is not correct?

We have now mapped our NAI-MaP reactivities to references that better fit the crystal structures and show that our reactivities are largely consistent with known paired and unpaired bases; using a reactivity cutoff of 0.5 as paired, we saw that 94% of our high reactivities (reactivity ≥ 0.5) fall on single-stranded bases in the model. We have now updated this data as Figure 1c

Figure 10. NAI-MaP reactivities mapped onto known 5' and 3' UTR structures in dengue 2 genome. A more intense red color indicates that the base has higher NAI-MaP reactivity. in the manuscript.

As most of the well-known structures are found only in 5' and 3' ends of the genomes, we also correlated our pair-wise interactions with other high throughput interactome datasets in Zika. We observed a high pearson correlation of $R=0.63$ and $R=0.62$ between our two replicates of SPLASH interactions with the COMRADES data presented in the Ziv O. et al paper³. In

Figure 11. 2 dimensional matrixes indicating the position of pair-wise RNA interactions in Zika Brazil strain, as identified by SPLASH (in blue) and by COMRADES (in red). SPLASH data shows good correlation with COMRADES data (Pearson, $R=0.625$).

contrast, the correlation drops to $R=0.36$ when we compared Ziv O's data to a shuffled control in our dataset. This high correlation suggests that our data reproducibly captures in cell interactions that are stably present across different labs, protocols, and experiments. We have now included this data as Extended data 9b in the manuscript.

In addition to the Ziv O. paper, we have also crosschecked the top interactions from Li. P et al and show that all of their top five interactions are abundantly captured in our data. This again indicates that our data is of high quality. In contrast to the recent structure papers on Zika, which have performed proximity mapping of a single Zika virus (Ziv. O et al) and of two Zika strains (Li. P et al)⁴, we performed structure mapping of four different Zika strains, reflecting the diversity of Zika viruses in the original African strain, the Brazil strain, French Polynesia strain, and the South East Asian Singapore strain. We believe that our data contains a rich repository of information, not only for studying the similarities, but also the differences, between the

Figure 11. SPLASH captures pair-wise interactions seen in other independent datasets. The blue box indicates the top five interactions identified in Li P et al. SPLASH identifies all five interactions at high read counts, indicating that our dataset is reproducible and robust.

strains.

5. The data dealing with unfolding of the RNA genome in the cell, and the role of the ribosome and the action of helicases is underdeveloped. To a certain degree, this is obvious – we know that the viral genome is the template for infection processes in the cell that must unwind it (replication and translation) and we know that it exists at some point as a fully double-stranded intermediate; hence the conclusion that “the structured state we observe in side cells is a result of active unwinding by enzymes...” and that the viral replicase machinery could play a roles not novel is a bit odd because we knew this...its settled virology. So, I'm left wondering: what did we learn here?

We thank the reviewer for his/her comments. The ability to map RNA structures inside cells in a high throughput manner is a fairly new technological development. As such, much remains to be learnt with regards to the structural dynamics of RNA structures in vivo versus in vitro and under different cellular states. RNA structures have been shown to be more open inside cells than in vitro, and a few groups have identified the ribosome as the major helicase that unwinds RNA structures inside cells^{5,6}. As we also observe that longer range interactions in dengue and Zika genomes tend to be disrupted in cells versus in vitro⁷, we tested the hypothesis that the ribosome is also the major helicase unwinding viral structures. Surprisingly, unlike mRNAs which gained structure when ribosomes are inhibited, dengue RNAs do not show significant structural changes upon ribosome inhibition. This suggests that while the ribosome is the major helicase acting on mRNAs inside cells, it is not the major helicase that is acting on dengue and Zika. We agree with the reviewer that while this is a potentially interesting observation, more work is needed to inhibit the helicase to dissect structural heterogeneity of viral structures, and that it is outside the scope of this manuscript. We have now removed the section on ribosome inhibition and dengue/Zika structure to streamline the manuscript.

6. Related to the above, infection by these RNA viruses is a very dynamic process in which the RNA plays several roles. In addition, these roles change over time and thus the amount of RNA in a given population (translating, replicating, packaging) also changes. In addition, these viruses generate large amount of several non-coding subgenomic RNAs that contain part of the genomic RNA sequence (I do not think this is even mentioned but it is a critical point!), hence there are at least 2-3 different RNA species that share sequence during infection. Each may have different structures and be doing different things. The authors do mention different structural states existing, but this is not fleshed out and thus I am left feeling like I don't really know much more about the reality of the structure of these RNAs in the cell than I did before. While the authors do not need to have all the answers, they need to talk about these data in light of these virological realities, otherwise it is superficial. This could be a paper in and of itself, if properly controlled and discussed.

We thank the reviewer for his/her comments. Due to short read sequencing, we are unable to distinguish between sRNA from the full length virus RNA. We have now discussed the complexities of the different structural states in the discussion section of the manuscript.

7. Overall, the discussion section leaves many mysterious and unanswered questions. I'll just say that two paragraphs is not nearly enough to cover all the caveats, various interpretations,

integration with known virology and RNA structure, etc. given the density and amount of data presented.

We thank the reviewer for his/her comments. We have now deepened the analysis of our manuscript and expanded our discussion to better integrate our data with known virology and to address various interpretations of our data.

Minor points

Other points:

1. Fig 1c+d: I think clustering what's <0.5 separately from what's >0.5 hides away information that would be best covered by a spectrum of three or more colors. I don't think that a reactivity of 0.49 for example would correspond to a paired residue, although here it would be black. And how many reactivities are in that 0.3-0.7 range anyway? Better to stick to the same scale as on Fig 2b.

We thank the reviewer for his/her comments. We have now removed discretization of the data and use the continuum of NAI-MaP reactivities directly in our analysis. We have updated Figure 1c, Figure 2b, Extended data 2c, 3b and 5a in the manuscript.

2. lines 125-127: "As expected, regions that share similar structure patterns between the 8 viruses generally share higher sequence identity, consistent with the idea that sequence plays an important role in RNA structure". It's pretty obvious that sequence is important for structure formation so this statement comes off as odd.

We thank the reviewer for his comments- as structure similarity tends to trend with sequence similarity, we used increased structure similarity over sequence similarity to identify structurally conserved elements. We have re-worded this in the manuscript.

3. line 146: Converting SHAPE reactivity to pseudo-energy terms has been shown by others to not be appropriate, unless that pseudo energy term is not regarded as an energy, but as a probabilistic inference. It's mostly useful in the context of such long RNAs when shown to support structures derived from other methods, such as sequence alignment, or probing using other probes (see Eddy SR, Annu Rev Biophys 43:433-456 (2014)).

We thank the reviewer for his/her comment. We calculated the distribution of SHAPE reactivity in our data for known 5' and 3' structural elements across all our systems and compared them to the analysis presented by Eddy SR, 2014 paper (Figure 6 in this rebuttal)¹. We observe distributions of reactivity for known and unknown bases congruent with what is presented by Eddy, namely that the most likely reactivity observed is low in case of paired and unpaired bases, but that the likelihood ratio for higher-reactivity bases indicates a high probability of unpairedness. We changed the wording of our manuscript to highlight that SHAPE reactivity is used for probabilistic inference of likely unpaired regions and added this analysis as Extended data 2f in the manuscript.

4. Fig 2b: I see many regions with problems in there: see 69-88, 782-802, 2247-2275, 4090-4120, either they are paired and should not be shown as unpaired, or vice-versa. What's going on here?

We thank the reviewer for his/her comments. NAI-MaP reactivities that do not fit modelled structures could be due to: 1) the natural distribution of NAI-MaP reactivities in paired and unpaired bases; and/or 2) the fact that bases exist in alternative conformations in the virions (Figure 7 in this rebuttal).

Reviewer #4 (Remarks to the Author):

The Authors have adequately addressed the concerns of Reviewer-1. Overall, the reworked manuscript provides several novel insights into the global genome organization of flaviviruses. Importantly, candidate long-range RNA-RNA interactions were tested for functionality, and were shown to be relevant. The study is well suited for the audience of this journal.

We thank the reviewer for his/her enthusiasm of the manuscript.

There are, however, a few issues that require attention prior to further consideration.

Line 47 – Abstract - change “the importance of these new” to “the importance of one of these new”, because only one interaction was assessed by compensatory mutational analysis

We thank the reviewer for his/her suggestion. We have now changed the wording in the abstract.

Line 78 – change “essential” to “important”, because the mutant viruses were still able to replicate to readily detectable levels

We thank the reviewer for his/her suggestion. We have now changed the wording in the manuscript.

Fig2b – explain in legend what is signified by the lighter shades of red, orange, and black in the key

We thank the reviewer for his/her suggestion. We have now followed reviewer 1’s suggestions to remove discretization in our data, and now present our data as a continuum of reactivities. A higher intensity of red on the scale in Figure 2b represents a higher NAI-MaP reactivity for that base, indicating that the base is more likely to be single stranded in the RNA.

Line 236 – “exact.....sequence identities across serotypes” – in Fig4d (upper and middle panels) green circles indicate covariation and thus substitutions within the interaction in different serotypes – therefore, “exact” sequence identities were not maintained for these two examples – thus, this statement needs correcting

We thank the reviewer for his/her suggestion. We have reworded the statement in the manuscript.

Fig5f,g,h,i – needs statistical evaluation – there are many very large error bars

We thank the reviewer for his/her suggestion. We have now performed statistical analysis and indicated the statistically significant changes with * in Figure 5 in the manuscript.

Fig5h has the results for the compensatory mutant (gold bars) in the graph, but it is MISSING results from the “mutant” (blue bars) containing substitutions in only the lower sequence in Fig5h - a critical piece of information.

We thank the reviewer for his/her suggestion. The mutant “blue bars” are actually not missing in our data. The three point mutations resulted in extremely severe attenuation such that we are unable to obtain live virus when we transfect the mutant virus into C6/36 mosquito cells in all four experiments attempted. In contrast, compensatory mutations partially rescue the extreme attenuation phenotype of the mutant. We have now performed six replicate experiments for this rescue and updated Figure 5g in the manuscript.

Reviewer #5 (Remarks to the Author):

I was asked to comment on the rebuttal to ref#2. I have the following comments:

General comments: I agree with ref#2 that the paper suffers from lack of functional studies. I do not think the authors provide sufficient new experiments to fulfil this shortcoming. The topic of long-range RNA interactions in viral genomes is interesting and worth studying. The RNA structure mapping in this paper is technically advanced and large amounts of relevant data are generated. The description is largely technical, sometimes difficult to follow, and I find the nice-looking Figures not easy to interpret. In my opinion where the paper falls short is, the interpretation of the results and subsequent experimentation focused on structure-function (very basic characterization of virus mutants). These limited functional studies are not very insightful. Recoded viral genomes may not only change RNA structures, but will also impact codon usage, codon bias, dinucleotide bias, all of which may affect replication in their own right. Vice versa, it is also possible to scramble the RNA of entire viral RNA genomes (while keeping codon usage and CpG bias the same) without major effects on replication – thus one could argue that the RNA structures in between the conserved UTRs are biologically not so important. For an example, see Fros et al., 2017 eLife. Without a better in-depth understanding of the biological importance of the ZIKV and DENV RNA structures, the paper is largely descriptive and it is not yet clear what these data mean for our understanding of the function of these RNA structures. It would have been beneficial if the authors, in addition to ZIKV and DENV, would have performed a control with a flavivirus that has known internal RNA structure, e.g. WNV with a ribosomal frameshift in the end of NS1, leading to NS1' (see Firth papers for these structures).

We thank the reviewer for his/her suggestions. To show that our data is accurate and robust, we have now correlated our pair-wise interaction data with two independent high throughput datasets that use different strategies to show that our data is robust and reproducible (Figures 11, 12 in our rebuttal). We have added this data as Extended data 9b in our manuscript.

Comment 1 and 2. I did not have sufficient experience with ribosome profiling to comment.

Comment 3. The structures have been shown as requested. The authors have also replaced the mutagenesis experiments with new experiments in which only point mutations disrupting RNA structures were tested. These viruses have been sequence verified. This is a better approach, but still does not provide the level of understanding that would be needed to propose that these RNA structures are indeed important and why. It is not entirely clear why some viruses are attenuated in vertebrate cells but not in invertebrate cells and vice versa. This at least would suggest that other mechanisms than RNA structure alone account for this difference, e.g. dinucleotide bias, which affects viruses in vertebrate but not invertebrate cells.

We thank the reviewer for his/her suggestions. To further validate that our newly detected pairwise RNA interactions do indeed exist in the genomes and that the mutations are disrupting the interactions, we have now performed pulldowns of wildtype and mutant interactions to assess the extent of interactions in the complementary strands by qPCR. Our experiments show that mutations result in a decreased amount of complementary strands being pulled down, supporting our results that the mutations indeed disrupt the pairing (Figure 9 in the rebuttal). We have added this data as Extended data 9c in the manuscript.

In addition, to further address the *in vivo* relevance of one of the mutants, we infected interferon deficient mice with equal titers of wildtype and mutant virus particles and checked for viremia two, four, and six days post infection. Mutant viruses showed a significant decrease in viremia four days post-infection, suggesting that the mutant is attenuated *in vivo*. We have now added this data as Figure 5i in the manuscript.

Figure 12. Point mutations that disrupt the pairing of the longer-range RNA interactions result in virus attenuation in Huh7 cells (Left) and in an interferon deficient mouse model (Right). Left: Wildtype and mutant viruses were used to infect Huh7 cells at an MOI =0.1 and the amount of virus that is produced 24 and 28hrs post infected is determined inside cells and in the supernatant by qPCR analysis. Less mutant viruses are produced in the cells and in the supernatant as compared to wildtype cells. Right: 2×10^5 PFU of wildtype and mutant viruses were injected into interferon deficient A129 mice. Virus titers in the blood serum were determined by plaque assays 2, 4, 6 days post-infection.

Comment 4. The authors responded satisfactorily to this comment.

References

1. Eddy, S. R. Computational analysis of conserved RNA secondary structure in transcriptomes and genomes. *Annu. Rev. Biophys.* **43**, 433-456 (2014).
2. Rivas, E., Clements, J. & Eddy, S. R. A statistical test for conserved RNA structure shows lack of evidence for structure in lncRNAs. *Nat. Methods* **14**, 45-48 (2017).
3. Ziv, O. *et al.* COMRADES determines in vivo RNA structures and interactions. *Nat. Methods* **15**, 785-788 (2018).
4. Li, P. *et al.* Integrative Analysis of Zika Virus Genome RNA Structure Reveals Critical Determinants of Viral Infectivity. *Cell. Host Microbe* **24**, 875-886.e5 (2018).
5. Beaudoin, J. D. *et al.* Analyses of mRNA structure dynamics identify embryonic gene regulatory programs. *Nat. Struct. Mol. Biol.* **25**, 677-686 (2018).
6. Rouskin, S., Zubradt, M., Washietl, S., Kellis, M. & Weissman, J. S. Genome-wide probing of RNA structure reveals active unfolding of mRNA structures in vivo. *Nature* (2013).
7. Mustoe, A. M. *et al.* Pervasive Regulatory Functions of mRNA Structure Revealed by High-Resolution SHAPE Probing. *Cell* **173**, 181-195.e18 (2018).

Reviewers' Comments:

Reviewer #4:

Remarks to the Author:

In this Reviewer's opinion, the Authors have adequately addressed the issues raised by the Referees.

Reviewer #6:

Remarks to the Author:

Dear authors: see my comments below to your response to Reviewer #3 inserted between *** symbols.

Specific Points:

1. The authors are quite confident about the "correctness" of their structure, and to a certain degree they are building on this term being applied in papers by others in the field. But this is dangerous because it gives the false impression that we now know what THE structure of this RNA is. In fact, there certainly are many different structures co-existing. The authors do mention that they see alternate structures for 40% of the RNA, this fact needs to be discussed and the data and resultant structures interpreted in light of this observation. I do not understand how the variation fits with the single 2D model shown in Figure 2.

We thank the reviewer for these insightful comments. We have now integrated both local secondary structure information and pair-wise RNA interactome information into structure modeling by using NAI-MaP reactivities as structural constraints to generate an initial ensemble of 1000 potential structures. We then selected the top 10 structures based on the best fit with our SPLASH interactome data. We confirmed that NAI-MaP constrained and SPLASH selected structures are significantly enriched for correct structures, based on 16S and 18S rRNA structures ($p < 0.003$). Top NAI-MaP constrained, SPLASH selected structures are present in the majority of the structural clusters, representing the diversity of structures present in dengue and Zika. We have now added this data as Extended data 8 in the manuscript.

***Stepping in for reviewer #3, I wish to thank the authors for addressing reviewer #3's concerns by adding these data to the manuscript. Yet, this further highlights that more deserves to be written about these alternative structures, especially as they make for such a high proportion of the RNA. This is also much of an uncharted territory in the current literature, but because as the authors mention they are probably among the first to correlate probing and SPLASH data to derive more meaningful structures, I think we would all benefit from directly looking at what these structures might look like. If they managed to do that, the authors would be planting a flag and set themselves as the new standard for analyzing structuromes.

I here second reviewer #3 in that the data shown should lead to at least two papers. My suggestion would be to have the first paper focus on the discovery of the structured elements and the long-range interactions, and the second focus on the alternative structures, or "structural heterogeneity" as the authors discuss it in the present manuscript. I would submit these papers for back-to-back publications. Currently, even with 5 multi-panel main text figures, 9 SI figures, and several tables, >95% of this huge body of data is lost to anyone not on the list of authors. For example on figure 4, I am left to wonder what similar sequences in DENV and ZIKA might pair differently, as the title of the figure suggests? Panels d and e are nice to show similarities, but what about differences? Some are shown on Figure 3, but there are clearly many more.

For example the data shown in Figure 2b only represents DENV-1, and it is overall difficult to grasp any detail at the scale of the figure. I need to zoom in at 900% on the PDF to see the details, but in any case the sequence is not shown, and only probing data for one of the stains is displayed, on only

one structure... It would make sense to also be able to access the alternative secondary structures derived by the authors (not the 1000 possible ones, but at least the top 10 hits the authors report), together with an overlay of the probing data, and that sort of thing. Computational resources and experts to make such data available to everyone are I am sure plentiful in Singapore! Naturally, this work may lead to yet a third paper to describe such a tool, similarly to when Phil Bevilacqua and Sarah Assmann published genome-wide probing data (<https://www.nature.com/articles/ncomms3971> , <https://academic.oup.com/bioinformatics/article/31/16/2668/321420>).***

For the sake of clarity in representing how dengue/Zika structures may look like inside virion particles, in Figure 2b in the manuscript we have presented the structure models for those with the best fit to known 3' and 5' UTRs in dengue and Zika. We have now clarified in the manuscript that this is just one potential structure model out of an ensemble. We do note that our 16 structurally conserved regions in DENV-1 are enriched for low Shannon entropies (Extended data 4d), suggesting that these elements are more likely to show unique structures within the ensembles, and hence may represent more stable structures.

Although it is very nice to have the complete overview of the genome displayed like this, it comes with shortcomings as I mentioned in my response to the previous point. I also wish to point out that no matter how careful the authors are at stating this is only one potential structure, the published structure is the one the reader will see, and therefore remember, work with, use as a reference, etc. Hence, more work should be done upstream on the authors' part, so that the alternative structures get equal focus, or at least are better able to judge why this structure would be better than the other ones. Even among predictions that might be less good according to computational analysis, there might be answers to key biological questions.

When they are talking about alternate structures, do they mean alternate long-distance interactions but conserved local 2D structures? Or is variation found in local region but not in long-range interactions? Or are they coupled? Are there some structures that are mutually exclusive between various alternate conformations?

We thank the reviewer for these comments. When we talk about alternate structures, we do not distinguish between short (<500 bases) or long (≥ 500 bases) pair-wise interactions. To test whether shorter- or longer-range interactions tend to be more variable and hence form more alternative structures, we calculated the proportion of short vs long interactions that are involved in alternative base pairing. We observed that both shorter and longer range interactions tend to form similar numbers of alternative structures, although longer interactions have a slight tendency to form more interactions (48.8% in >500 bases vs 41.6% in <500 bases, $p=0.12$, Figure 3c in the manuscript). Interestingly, we noticed that in cell interactions tend to be either unique or have only one alternative interaction (7.8% of in cell interactions form more than one alternate structure and 14.8% of in virion interactions form more than one alternate structure, $p=0.02$), while in virion interactions tend to show greater heterogeneity, suggesting that the genomes exist in a greater diversity of conformations inside virions.

In line with the comment from Reviewer #3, it would be nice to see a more systematic analysis of whether for example the 16 or 12 structured regions are maintained in the various alternative structures. Or when those are disrupted, which ones are actually disrupted, and what does the disruption look like? We might be able to grasp patterns for key aspects of virology by contrasting these various states.

To test whether shorter or longer interactions tend to be more conserved, and are hence shared across two or more viruses, we calculated the fraction of long vs short interactions that are shared in

>2 dengue or Zika strains. Interestingly, we observed that the shorter (<500 bases) interactions tend to be more conserved across viruses. We have now added this data as Extended data 7a in the manuscript.

Similarly, we also observed that shorter interactions inside virions are preferentially conserved in cells, indicating that shorter interactions are less variable.

In the 2D model of Figure 2, how should I think about chemical probing that does NOT fit with the 2D model (easily spotted)? Is this indicative of alternate structures? This is all unclear because the interpretation and discussion is very shallow.

We thank the reviewer for these comments. The NAI-MaP reactivity reflects the probability that a base is paired or unpaired, but does not strictly indicate structuredness. As such, not all of the unpaired bases will have high reactivity, such that the mode of SHAPE-reactivity for unpaired bases is still at 01. We plotted the distribution of our reactivity in known paired and unpaired bases in the 5' and 3'UTRs of dengue and Zika viruses and saw that known unpaired bases tend to have higher reactivity profiles, as expected from the Eddy SR 2014 paper. We have now added this data as Extended data 2f in the manuscript.

Unfortunately the discrepancy between certain reactivities and their paired/unpaired state in the secondary structure model is an inherent limitation of this kind of method for predicting structures. By referring to Eddy's work I think the authors have done the best they could to address that concern.

To determine whether chemical reactivities that do not fit the structure model could be a result of alternate pairings, we calculated the distribution of Shannon entropies for bases that fit the model, versus the bases that do not, as bases with low Shannon entropies are indicative of unique structures. Interestingly, we observe that bases that do not fit the model tend to have higher Shannon entropies, suggesting that they may be involved in alternate pairings. We have now added this data as Extended data 5c.

***This and the previous point further support the need to be able to look at the reactivity for the same nucleotides in various structures. Indeed, a reactive residue that might be paired in the number 1 hit structure, might be unpaired in another, which would tell us something about alternate conformations. In turn, being able to contrast such data might help computational biologists to better make use of probing data to predict meaningful alternative structures.

As the authors write in their discussion (line 345), functional significance still remains to be tested experimentally. At first, upon publication, most likely virologists in the field will want to be able to dive into the data to see whether it supports a particular motif, interaction or region they have been focussing on. But in order to do that, they need to be able to readily access such data! (it can be expected that most won't be computational biologists).***

2. Moving from the global model to specific interactions or structured elements, there is again a need for more detailed analysis and discussion. At a minimum, for each new element identified, it is important to show whether the newly proposed structure is supported by sequence alignments. There is a rich repository of sequence data from viral isolates of all kinds of flaviviruses which could readily be used. (hundreds from dengue alone). To be fair, this is shown to some extent on Fig 2b, but there is it mostly the 3'UTR that's convincing (which we already know a lot about). I am missing the alignments needed for me to really assess the validity of the reported covarying base pairs. Apparent covariation can occur in random sequences; thus, thus many sequences are needed to assess this, and that analysis should be present somewhere for me to judge. Conversely, if these regions of the RNA are too highly conserved for much covariation, then the reader needs to be able to see how conserved. It looks like they have access to all that information (extended data 3 and 4).

We thank the reviewer for his/her thoughtful comments. We have now performed covariation analysis using ~5000 sequences from dengue and Zika genomes, using the program R-Scape2. Our NAI-MaP constrained structure models are enriched for co-varied bases (418-450 covaried bases for NAI-MaP

constrained structures vs 258 covaried bases in the shuffled structure), as compared to shuffled sequences, providing support for the modelled structures. We have now provided an example of co-varied bases in a newly identified alternative pair-wise RNA interaction (from Figure 3) and have now uploaded the alignments as a supplementary data file 1.

R-scape is the new gold standard for validating sequence alignment data, so the authors are again demonstrating they are doing the best to make such data meaningful and available. Here again though, it would be nice to have the alignment as a supplementary file, together with the R-scape output (2D structures, plots and E-values in particular). It is possible that locally some of the alternate structures may support more covariations. This may, again, be a paper in itself!

3. There is a real need to verify that the mutations made have the structural effect that is predicted. When making mutations to alter RNA structure, it is the accepted practice in the field to first assess that the mutations have the predicted or intended effect by repeating the structural or biophysical analysis (in this case, probing) with the mutant. If one "side" of a long-range interaction is altered, then probing should show a change in the pattern there and in the other "side" of the interaction. This validates the interaction and the mutation and allows the functional effects of the mutation to be properly interpreted. These are essential controls.

We thank the reviewer for these comments. We have sequenced the mutants by capillary sequencing to make sure that the mutations are correct in Zika viruses. In addition, we have also performed an experiment in which, if one "side" of a long range interaction is mutated, the other "side" will not be able to interact as well. As such, if we fragment the Zika genome, pull down one side of the mutated interaction, and qPCR for the presence of the other side, we may determine whether the mutation has disrupted the interaction. Our experiments show that upon fragmentation, qPCR of one mutant strand results in the poorer pull down of the interacting strand, validating that the pair-wise RNA interactions indeed exist in the host cells and that the mutations disrupt the pair-wise interaction as expected. We have now added this data as Extended data 9c in the manuscript.

Although the pull down experiment proposed by the authors indirectly addresses the question of reviewer #3, it does not directly demonstrate that reactivities increase where the mutations have been made, which was the control reviewer #3 asked for.

4. Another validation tool could be to calculate correlation coefficients between the predicted structures and those that we already know of based on prior studies (especially for the 3' UTR). That would give a confidence criteria for sequences elsewhere on the genome for which we had no prior reference.

We thank the reviewer for his/her comments. We have previously shown that integrating NAI-MaP reactivities into structure modeling improves the accuracy of our modelled 16S and 23S rRNA structures (Extended data 5b). We have now revised our predicted structure models by optimizing the parameters used for including SHAPE data into the structure modeling procedure and generating an initial ensemble of 1000 potential structures. The structures with the best concordance to 5'/3' UTRs show 91%, 92%, 78% and 91% accuracy (modelled dengue1-4 structures respectively) compared to the known dengue 2 5'UTR structure, and an accuracy of 76%, 81%, 78% and 85% (for modelled Zika Africa, Brazil, French Polynesia and Singapore strains, respectively) compared to the Zika Brazil 5'UTR structure. We see a concordance of 87%, 93%, 86% and 63% accuracy (modelled dengue1-4 structures respectively) to the known dengue 2 3'UTR structures, and 73%, 84%, 78%, 83% accuracy (Zika Africa, Brazil, French Polynesia and Singapore respectively), compared to the Zika Brazil 3'UTR structure. As most of the known structures are focused on dengue 2 virus and the Brazil and French Polynesia strain of Zika, lower similarities between predicted strains and known structures could also be due to differences in actual structures between the viruses. We have now included this information in the manuscript.

***I agree that correlation (a term more often employed than 'concordance' for this kind of analysis)

between the new data and what we already know about the structured elements of these viruses is essential to our understanding. This should be one of the key points in the paper #1 that would be focused on structures. Although the authors mention these values (no need for two decimals by the way – see lines 155-168), it would make sense to incorporate them into figures, and again, use them to better contrast various alternative possibilities for folding these RNAs.***

Related, I am concerned that the authors present the secondary structure of the 3' UTR for reference, but parts of this 2D structure are not correct based on published, validated structures (including 3-D crystal structures!). If this is not correct, what else is not correct?

We have now mapped our NAI-MaP reactivities to references that better fit the crystal structures and show that our reactivities are largely consistent with known paired and unpaired bases; using a reactivity cutoff of 0.5 as paired, we saw that 94% of our high reactivities (reactivity ≥ 0.5) fall on single-stranded bases in the model. We have now updated this data as Figure 1c in the manuscript.

This point could be further addressed by making it visually explicit that structures derived from probing are similar to those derived by other means, for all structured element for which there is prior structural information.

As most of the well-known structures are found only in 5' and 3' ends of the genomes, we also correlated our pair-wise interactions with other high throughput interactome datasets in Zika. We observed a high Pearson correlation of $R=0.63$ and $R=0.62$ between our two replicates of SPLASH interactions with the COMRADES data presented in the Ziv O. et al paper³. In contrast, the correlation drops to $R=0.36$ when we compared Ziv O's data to a shuffled control in our dataset. This high correlation suggests that our data reproducibly captures in cell interactions that are stably present across different labs, protocols, and experiments. We have now included this data as Extended data 9b in the manuscript.

An $R=0.63$ is equivalent to an R^2 of 0.4. Although the correlation is OK for such a large dataset, there are still a lot of differences. What do those correspond to precisely? Could those be mapped? Again, as with explicitly contrasting alternate structures, this could reveal structural information that could help understand the biology.

In addition to the Ziv O. paper, we have also crosschecked the top interactions from Li. P et al and show that all of their top five interactions are abundantly captured in our data. This again indicates that our data is of high quality. In contrast to the recent structure papers on Zika, which have performed proximity mapping of a single Zika virus (Ziv. O et al) and of two Zika strains (Li. P et al)⁴, we performed structure mapping of four different Zika strains, reflecting the diversity of Zika viruses in the original African strain, the Brazil strain, French Polynesia strain, and the South East Asian Singapore strain. We believe that our data contains a rich repository of information, not only for studying the similarities, but also the differences, between the strains.

"We believe that our data contains a rich repository of information": ABSOLUTELY! which is why it is crucial to properly present and deliver the load of information to the large community of scientists that work on such viruses. Especially since there is little doubt that the data are of high quality.

5. The data dealing with unfolding of the RNA genome in the cell, and the role of the ribosome and the action of helicases is underdeveloped. To a certain degree, this is obvious – we know that the viral genome is the template for infection processes in the cell that must unwind it (replication and translation) and we know that it exists at some point as a fully double-stranded intermediate; hence the conclusion that "the structured state we observe in side cells is a result of active unwinding by enzymes..." and that the viral replicase machinery could play a role not novel is a bit odd because we knew this...its settled virology. So, I'm left wondering: what did we learn here?

We thank the reviewer for his/her comments. The ability to map RNA structures inside cells in a high

throughput manner is a fairly new technological development. As such, much remains to be learnt with regards to the structural dynamics of RNA structures in vivo versus in vitro and under different cellular states. RNA structures have been shown to be more open inside cells than in vitro, and a few groups have identified the ribosome as the major helicase that unwinds RNA structures inside cells^{5, 6}. As we also observe that longer range interactions in dengue and Zika genomes tend to be disrupted in cells versus in vitro⁷, we tested the hypothesis that the ribosome is also the major helicase unwinding viral structures. Surprisingly, unlike mRNAs which gained structure when ribosomes are inhibited, dengue RNAs do not show significant structural changes upon ribosome inhibition. This suggests that while the ribosome is the major helicase acting on mRNAs inside cells, it is not the major helicase that is acting on dengue and Zika. We agree with the reviewer that while this is a potentially interesting observation, more work is needed to inhibit the helicase to dissect structural heterogeneity of viral structures, and that it is outside the scope of this manuscript. We have now removed the section on ribosome inhibition and dengue/Zika structure to streamline the manuscript.

I support the authors's decision to remove this aspect from the current manuscript.

6. Related to the above, infection by these RNA viruses is a very dynamic process in which the RNA plays several roles. In addition, these roles change over time and thus the amount of RNA in a given population (translating, replicating, packaging) also changes. In addition, these viruses generate large amount of several non-coding subgenomic RNAs that contain part of the genomic RNA sequence (I do not think this is even mentioned but it is a critical point!), hence there are at least 2-3 different RNA species that share sequence during infection. Each may have different structures and be doing different things. The authors do mention different structural states existing, but this is not fleshed out and thus I am left feeling like I don't really know much more about the reality of the structure of these RNAs in the cell than I did before. While the authors do not need to have all the answers, they need to talk about these data in light of these virological realities, otherwise it is superficial. This could be a paper in and of itself, if properly controlled and discussed.

We thank the reviewer for his/her comments. Due to short read sequencing, we are unable to distinguish between sfRNA from the full length virus RNA. We have now discussed the complexities of the different structural states in the discussion section of the manuscript.

This aspect makes it even more essential to discuss alternative structures in a completely separate manuscript. Currently, the probing data is a blur of several RNA species of various lengths, which may adopt different structures, as reviewer #3 points out. It is important to be able to sort out whether the alternative structures are associated to different states of the same RNA, or to different RNAs with a similar sequence.

7. Overall, the discussion section leaves many mysterious and unanswered questions. I'll just say that two paragraphs is not nearly enough to cover all the caveats, various interpretations, integration with known virology and RNA structure, etc. given the density and amount of data presented.

We thank the reviewer for his/her comments. We have now deepened the analysis of our manuscript and expanded our discussion to better integrate our data with known virology and to address various interpretations of our data.

Although the discussion appears to now be of a sufficient length for one manuscript, the only way to expand it would be to continue it in... additional papers.

Minor points

Other points:

1. Fig 1c+d: I think clustering what's <0.5 separately from what's >0.5 hides away information that would be best covered by a spectrum of three or more colors. I don't think that a reactivity of 0.49 for example would correspond to a paired residue, although here it would be black. And how many reactivities are in that 0.3-0.7 range anyway? Better to stick to the same scale as on Fig 2b.

We thank the reviewer for his/her comments. We have now removed discretization of the data and use the continuum of NAI-MaP reactivities directly in our analysis. We have updated Figure 1c, Figure 2b, Extended data 2c, 3b and 5a in the manuscript.

This point has been properly addressed.

2. lines 125-127: "As expected, regions that share similar structure patterns between the 8 viruses generally share higher sequence identity, consistent with the idea that sequence plays an important role in RNA structure". It's pretty obvious that sequence is important for structure formation so this statement comes off as odd.

We thank the reviewer for his comments- as structure similarity tends to trend with sequence similarity, we used increased structure similarity over sequence similarity to identify structurally conserved elements. We have re-worded this in the manuscript.

This point has been properly addressed.

3. line 146: Converting SHAPE reactivity to pseudo-energy terms has been shown by others to not be appropriate, unless that pseudo energy term is not regarded as an energy, but as a probabilistic inference. It's mostly useful in the context of such long RNAs when shown to support structures derived from other methods, such as sequence alignment, or probing using other probes (see Eddy SR, Annu Rev Biophys 43:433-456 (2014)).

We thank the reviewer for his/her comment. We calculated the distribution of SHAPE reactivity in our data for known 5' and 3' structural elements across all our systems and compared them to the analysis presented by Eddy SR, 2014 paper (Figure 6 in this rebuttal)¹. We observe distributions of reactivity for known and unknown bases congruent with what is presented by Eddy, namely that the most likely reactivity observed is low in case of paired and unpaired bases, but that the likelihood ratio for higher-reactivity bases indicates a high probability of unpairedness. We changed the wording of our manuscript to highlight that SHAPE reactivity is used for probabilistic inference of likely unpaired regions and added this analysis as Extended data 2f in the manuscript.

This point has been properly addressed.

4. Fig 2b: I see many regions with problems in there: see 69-88, 782-802, 2247-2275, 4090-4120, either they are paired and should not be shown as unpaired, or vice-versa. What's going on here?

We thank the reviewer for his/her comments. NAI-MaP reactivities that do not fit modelled structures could be due to: 1) the natural distribution of NAI-MaP reactivities in paired and unpaired bases; and/or 2) the fact that bases exist in alternative conformations in the virions (Figure 7 in this rebuttal).

These examples constitute further reasons for presenting and discussing alternative structures in more details!

We thank the reviewers for their comments, which has helped to make the data more accessible to the readers.

REVIEWERS' COMMENTS:

Reviewer #4 (Remarks to the Author):

In this Reviewer's opinion, the Authors have adequately addressed the issues raised by the Referees.

Reviewer #6 (Remarks to the Author):

Dear authors: see my comments below to your response to Reviewer #3 inserted between *** symbols.

Specific Points:

1. The authors are quite confident about the "correctness" of their structure, and to a certain degree they are building on this term being applied in papers by others in the field. But this is dangerous because it gives the false impression that we now know what THE structure of this RNA is. In fact, there certainly are many different structures co-existing. The authors do mention that they see alternate structures for 40% of the RNA, this fact needs to be discussed and the data and resultant structures interpreted in light of this observation. I do not understand how the variation fits with the single 2D model shown in Figure 2.

We thank the reviewer for these insightful comments. We have now integrated both local secondary structure information and pair-wise RNA interactome information into structure modeling by using NAI-MaP reactivities as structural constraints to generate an initial ensemble of 1000 potential structures. We then selected the top 10 structures based on the best fit with our SPLASH interactome data. We confirmed that NAI-MaP constrained and SPLASH selected structures are significantly enriched for correct structures, based on 16S and 18S rRNA structures ($p < 0.003$). Top NAI-MaP constrained, SPLASH selected structures are present in the majority of the structural clusters, representing the diversity of structures present in dengue and Zika. We have now added this data as Extended data 8 in the manuscript.

***Stepping in for reviewer #3, I wish to thank the authors for addressing reviewer #3's concerns by adding these data to the manuscript. Yet, this further highlights that more deserves to be written about these alternative structures, especially as they make for such a high proportion of the RNA. This is also much of an uncharted territory in the current literature, but because as the authors mention they are probably among the first to correlate probing and SPLASH data to derive more meaningful structures, I think we would all benefit from directly looking at what these structures might look like. If they managed to do that, the authors would be planting a flag and set themselves as the new standard for analyzing structuromes.

We thank the reviewer for his/her enthusiasm. We have now included full secondary structure models for three structures chosen according to different criteria in the supplementary information (best SPLASH match, best 5'/3' prediction accuracy, 2nd best SPLASH after removing best SPLASH). As these secondary structure models take up a lot of space, we have also provided the top 10 best fitting structures (constrained by NAI-N3 and selected by SPLASH) in the form of arc plots (Supplementary Figures 11-16).

I here second reviewer #3 in that the data shown should lead to at least two papers. My suggestion would be to have the first paper focus on the discovery of the structured elements and the long-range interactions, and the second focus on the alternative structures, or "structural heterogeneity" as the authors discuss it in the present manuscript. I would submit these papers for back-to-back publications. Currently, even with 5 multi-panel main text figures, 9 SI figures, and several tables, >95% of this huge body of data is lost to anyone not on the list of authors. For example on figure 4, I am left to wonder what similar sequences in DENV and ZIKA might pair differently, as the title of the figure suggests? Panels d and e are nice to show similarities, but what about differences? Some are shown on Figure 3, but there are clearly many more.

We thank the reviewer for his/her suggestions. We have listed the identities of pair-wise interactions in Supplementary Data 2,3,8 and 9. To make the information more accessible, we have now also included sequence information for these interactions, so that it is easier for readers to examine the sequences.

For example the data shown in Figure 2b only represents DENV-1, and it is overall difficult to grasp any detail at the scale of the figure. I need to zoom in at 900% on the PDF to see the details, but in any case the sequence is not shown, and only probing data for one of the stains is displayed, on only one structure... It would make sense to also be able to access the alternative secondary structures derived by the authors (not the 1000 possible ones, but at least the top 10 hits the authors report), together with an overlay of the probing data, and that sort of thing. Computational resources and experts to make such data available to everyone are I am sure plentiful in Singapore!

Naturally, this work may lead to yet a third paper to describe such a tool, similarly to when Phil Bevilacqua and Sarah Assmann published genome-wide probing data (<https://www.nature.com/articles/ncomms3971> , <https://academic.oup.com/bioinformatics/article/31/16/2668/321420>).

We thank the reviewer for his/her comments. We have now added the sequence information back to the secondary structure models and also included structure models for 3 structures selected by SPLASH and best fit to known UTR structures. We have also included arc plots for the top 10 selected structures.

For the sake of clarity in representing how dengue/Zika structures may look like inside virion particles, in Figure 2b in the manuscript we have presented the structure models for those with the best fit to known 3' and 5' UTRs in dengue and Zika. We have now clarified in the manuscript that this is just one potential structure model out of an ensemble. We do note that our 16 structurally conserved regions in DENV-1 are enriched for low Shannon entropies (Extended data 4d), suggesting that these elements are more likely to show unique structures within the ensembles, and hence may represent more stable structures.

Although it is very nice to have the complete overview of the genome displayed like this, it comes with shortcomings as I mentioned in my response to the previous point. I also wish to point out that no matter how careful the authors are at stating this is only one potential structure, the published structure is the one the reader will see, and therefore remember, work with, use as a reference, etc. Hence, more work should be done upstream on the authors' part, so that the alternative structures get equal focus, or at least are better able to judge why this structure would be better than the other ones. Even among predictions that might be less good according to computational analysis, there might be answers to key biological questions.

We thank the reviewer for his/her comments. We have now included structure models for 3 structures selected by SPLASH and best fit to known UTR structures. We have also included arc plots for the top 10 selected structures to illustrate the complexity of the dengue and Zika structures we observe.

When they are talking about alternate structures, do they mean alternate long-distance interactions but conserved local 2D structures? Or is variation found in local region but not in long-range interactions? Or are they coupled? Are there some structures that are mutually exclusive between various alternate conformations?

We thank the reviewer for these comments. When we talk about alternate structures, we do not distinguish between short (<500 bases) or long (>=500 bases) pair-wise interactions. To test whether shorter- or longer-range interactions tend to be more variable and hence form more alternative structures, we calculated the proportion of short vs long interactions that are involved in alternative base pairing. We observed that both shorter and longer range interactions tend to form similar numbers of alternative structures, although longer interactions have a slight tendency to form more interactions (48.8% in >500bases vs 41.6% in <500 bases, $p=0.12$, Figure 3c in the manuscript). Interestingly, we noticed that in cell interactions tend to be either unique or have only one alternative interaction (7.8% of in cell interactions form more than one alternate structure and 14.8% of in virion interactions form more than one alternate structure, $p=0.02$), while in virion interactions tend to show greater heterogeneity, suggesting that the genomes exist in a greater diversity of conformations inside virions.

In line with the comment from Reviewer #3, it would be nice to see a more systematic analysis of whether for example the 16 or 12 structured regions are maintained in the various alternative structures. Or when those are disrupted, which ones are actually disrupted, and what does the disruption look like? We might be able to grasp patterns for key aspects of virology by contrasting these various states.

We thank the reviewer for his/her comments. Structures with low Shannon entropies tend to form unique structures (maintained in various alternative structures) and are likely to be less variable than structures with high Shannon. We

Figure 1. Distribution of Shannon entropies in nominated dengue and Zika structures. Violin plots showing distribution of Shannon entropies in the selected 16 and 12 regions in dengue and Zika, versus all regions in the genome. P -value is calculated by Wilcoxon Ranked Sum Test. The lines on the violin plot indicate maximum, median and minimum values.

calculated the distribution of Shannon entropies in the 16 and 12 structured regions in dengue and Zika respectively and showed that these regions have significantly lower Shannon entropies, suggesting that their structures are more likely to be maintained (Supplementary Figure 5d). We have also highlighted these 16 and 12 structures in the alternative secondary structure models (Supplementary Figures 11-16) for ease of comparison.

To test whether shorter or longer interactions tend to be more conserved, and are hence shared across two or more viruses, we calculated the fraction of long vs short interactions that are shared in >2 dengue or Zika strains. Interestingly, we observed that the shorter (<500 bases) interactions tend to be more conserved across viruses. We have now added this data as Extended data 7a in the manuscript.

Similarly, we also observed that shorter interactions inside virions are preferentially conserved in cells, indicating that shorter interactions are less variable.

In the 2D model of Figure 2, how should I think about chemical probing that does NOT fit with the 2D model (easily spotted)? Is this indicative of alternate structures? This is all unclear because the interpretation and discussion is very shallow.

We thank the reviewer for these comments. The NAI-MaP reactivity reflects the probability that a base is paired or unpaired, but does not strictly indicate structuredness. As such, not all of the unpaired bases will have high reactivity, such that the mode of SHAPE-reactivity for unpaired bases is still at 01. We plotted the distribution of our reactivity in known paired and unpaired bases in the 5' and 3'UTRs of dengue and Zika viruses and saw that known unpaired bases tend to have higher reactivity profiles, as expected from the Eddy SR 2014 paper. We have now added this data as Extended data 2f in the manuscript.

Unfortunately the discrepancy between certain reactivities and their paired/unpaired state in the secondary structure model is an inherent limitation of this kind of method for predicting structures. By referring to Eddy's work I think the authors have done the best they could to address that concern.

We thank the reviewer for his comments.

To determine whether chemical reactivities that do not fit the structure model could be a result of alternate pairings, we calculated the distribution of Shannon entropies for bases that fit the model, versus the bases that do not, as bases with low Shannon entropies are indicative of unique structures. Interestingly, we observe that bases that do not fit the model tend to have higher Shannon entropies, suggesting that they may be involved in alternate pairings. We have now added this data as Extended data 5c.

***This and the previous point further support the need to be able to look at the reactivity for the same nucleotides in various structures. Indeed, a reactive residue that might be paired in the number 1 hit structure, might be unpaired in another, which would tell us something about alternate conformations. In turn, being able to contrast such data might help computational biologists to better make use of probing data to predict meaningful alternative structures.

As the authors write in their discussion (line 345), functional significance still remains to be tested experimentally. At first, upon publication, most likely virologists in the field will want to be able to dive into the data to see whether it supports a particular motif, interaction or region they have been focussing on. But in order to do that, they need to be able to readily access such data! (it can be expected that most won't be computational biologists).***

We thank the reviewer for his/her comments. We have provided per base NAI-MaP reactivity data for each virus as Supplementary data 1 and 2. The problem of structural heterogeneity is a very interesting one. Unfortunately, as we are unable to do single molecule structure probing, currently NAI-MaP reactivities at each base is a result of an average of reactivities of the base across all potential conformations. In contrast, SPLASH data can potentially dissect structural heterogeneity as a base along an RNA can only physically pair in one location. As such we have primarily focused on using SPLASH and a combination of NAI-MaP constrained and SPLASH selected information to study RNA structures in this study.

2. Moving from the global model to specific interactions or structured elements, there is again a need for more detailed analysis and discussion. At a minimum, for each new element identified, it is important to show whether the newly proposed structure is supported by sequence alignments. There is a rich repository of sequence data from viral isolates of all kinds of flaviviruses which could readily be used. (hundreds from dengue alone). To be fair, this is shown to some extent on Fig 2b, but there is it mostly the 3'UTR that's convincing (which we already know a lot about). I am missing the alignments needed for me to really assess the validity of the reported covarying base pairs. Apparent covariation can occur in random sequences; thus, thus many sequences are needed to assess this, and that analysis should be present somewhere for me to judge. Conversely, if these regions of the RNA are too highly conserved for much covariation, then the reader needs to be able to see how conserved. It looks like they have access to all that information (extended data 3 and 4).

We thank the reviewer for his/her thoughtful comments. We have now performed covariation analysis using ~5000 sequences from dengue and Zika genomes, using the program R-Scape2. Our NAI-MaP constrained structure models are enriched for co-varied bases (418-450 covaried bases for NAI-MaP constrained structures vs 258 covaried bases in the shuffled structure), as compared to shuffled sequences, providing support for the modelled structures. We have now provided an example of co-varied bases in a newly identified alternative pair-wise RNA interaction (from Figure 3) and have now uploaded the alignments as a supplementary data file 1.

***R-scape is the new gold standard for validating sequence alignment data, so the authors are again demonstrating they are doing the best to make such data meaningful and available. Here again though, it would be nice to have the alignment as a supplementary file, together with the R-scape output (2D structures, plots and E-values in particular).

It is possible that locally some of the alternate structures may support more covariations. This may, again, be a paper in itself!***

We thank the reviewer for his comments. We have now included the alignments of the ~5000 full-genome sequences used in our analysis as Supplementary data file 6-7. We have also included R-scape outputs for the three of the modelled alternative dengue and Zika structures. We included information on the interacting bases and the E values in the R-scape output for the ease of looking at the data. Observed co-varied bases are colored as green in the secondary structure models.

3. There is a real need to verify that the mutations made have the structural effect that is predicted. When making mutations to alter RNA structure, it is the accepted practice in the field to first assess that the mutations have the predicted or intended effect by repeating the structural or biophysical analysis (in this case, probing) with the mutant. If one "side" of a long-range interaction is altered, then probing should show a change in the pattern there and in the other "side" of the interaction. This validates the interaction and the mutation and allows the functional effects of the mutation to be properly interpreted. These are essential controls.

We thank the reviewer for these comments. We have sequenced the mutants by capillary sequencing to make sure that the mutations are correct in Zika viruses. In addition, we have also performed an experiment in which, if one "side" of a long range interaction is mutated, the other "side" will not be able to interact as well. As such, if we fragment the Zika genome, pull down one side of the mutated interaction, and qPCR for the presence of the other side, we may determine whether the mutation has disrupted the interaction. Our experiments show that upon fragmentation, qPCR of one mutant strand results in the poorer pull down of the interacting strand, validating that the pair-wise RNA interactions indeed exist in the host cells and that the mutations disrupt the pair-wise interaction as expected. We have now added this data as Extended data 9c in the manuscript.

Although the pull down experiment proposed by the authors indirectly addresses the question of reviewer #3, it does not directly demonstrate that reactivities increase where the mutations have been made, which was the control reviewer #3 asked for.

We thank the reviewer for his comments. As NAI-MaP reactivity is an average reactivity of a specific base across all conformations, a base that is now single-stranded because of mutations could potentially pair locally within itself, making the structural differences due to the mutations appear to be less pronounced. As such, we performed pulldown experiments to determine whether our mutations indeed disrupt the physical interactions that drive the base-pairing. Unfortunately, repeating the SPLASH experiment for all possible mutants is presently prohibitively expensive, and hence we used the described pull-down experiment to verify the interactions.

4. Another validation tool could be to calculate correlation coefficients between the predicted structures and those that we already know of based on prior studies (especially for the 3' UTR). That would give a confidence criteria for sequences elsewhere on the genome for which we had no prior reference.

We thank the reviewer for his/her comments. We have previously shown that integrating NAI-MaP reactivities into structure modeling improves the accuracy of our modelled 16S and 23S rRNA structures (Extended data 5b). We have now revised our predicted structure models by optimizing the parameters used for including SHAPE data into the structure modeling procedure and generating an initial ensemble of 1000 potential structures. The structures with the best concordance to 5'/3' UTRs show 91%, 92%, 78% and 91% accuracy (modelled dengue1-4 structures respectively) compared to the known dengue 2 5'UTR structure, and an accuracy of 76%, 81%, 78% and 85% (for modelled Zika Africa, Brazil, French Polynesia and Singapore strains, respectively) compared to the Zika Brazil 5'UTR structure. We see a concordance of 87%, 93%, 86% and 63% accuracy (modelled dengue1-4 structures respectively) to the known dengue 2 3'UTR structures, and 73%, 84%, 78%, 83% accuracy (Zika Africa, Brazil, French Polynesia and Singapore respectively), compared to the Zika Brazil 3'UTR structure. As most of the known structures are focused on dengue 2 virus and the Brazil and French Polynesia strain of Zika, lower similarities between predicted strains and known structures could also be due to differences in actual structures between the viruses. We have now included this information in the manuscript.

I agree that correlation (a term more often employed than 'concordance' for this kind of analysis) between the new data and what we already know about the structured elements of these viruses is essential to our understanding. This should be one of the key points in the paper #1 that would be focused on structures. Although the authors mention these values (no need for two decimals by the way – see lines 155-168), it would make sense to incorporate them into figures, and again, use them to better contrast various alternative possibilities for folding these RNAs.

We thank the reviewer for his/her suggestions. We have now included our NAI-MaP constrained models for the 5' and 3'UTRs of 4 dengue and 4 Zika strains in Supplementary Figure 7. This will allow readers to examine our modelled structures versus known structures in the reference.

Related, I am concerned that the authors present the secondary structure of the 3' UTR for reference, but parts of this 2D structure are not correct based on published, validated structures (including 3-D crystal structures!). If this is not

correct, what else is not correct?

We have now mapped our NAI-MaP reactivities to references that better fit the crystal structures and show that our reactivities are largely consistent with known paired and unpaired bases; using a reactivity cutoff of 0.5 as paired, we saw that 94% of our high reactivities (reactivity ≥ 0.5) fall on single-stranded bases in the model. We have now updated this data as Figure 1c in the manuscript.

This point could be further addressed by making it visually explicit that structures derived from probing are similar to those derived by other means, for all structured element for which there is prior structural information.

We thank the reviewer for his/her comments. We have now provided NAI-MaP reactivities mapped onto known references for all 4 dengue and Zika viruses in Supplementary Figure 2, so that the readers can better discern similarities and differences between our NAI-MaP reactivities and reference UTR structures.

As most of the well-known structures are found only in 5' and 3' ends of the genomes, we also correlated our pairwise interactions with other high throughput interactome datasets in Zika. We observed a high Pearson correlation of $R=0.63$ and $R=0.62$ between our two replicates of SPLASH interactions with the COMRADES data presented in the Ziv O. et al paper³. In contrast, the correlation drops to $R=0.36$ when we compared Ziv O's data to a shuffled control in our dataset. This high correlation suggests that our data reproducibly captures in cell interactions that are stably present across different labs, protocols, and experiments. We have now included this data as Extended data 9b in the manuscript.

An $R=0.63$ is equivalent to an R^2 of 0.4. Although the correlation is OK for such a large dataset, there are still a lot of differences. What do those correspond to precisely? Could those be mapped? Again, as with explicitly contrasting alternate structures, this could reveal structural information that could help understand the biology.

We thank the reviewer for his/her comments. We have tried to examine potential reasons behind differences between our data and Miska's data by 1) correlating both short-range and long range interactions with the Miska data, and 2) correlating abundant versus all interactions in both datasets (by excluding weak signals at the bottom 10% of either data set). However, these did not result in either an increased or decreased correlation between the two datasets, suggesting that the datasets are not biased by abundance or distance of interactions. The overlap of the data sets is presented in Supplementary Figure 17b. Further analysis beyond the scope of this publication is needed to fully understand the divergence of the two data sets, which may be caused by slight differences in methodology.

In addition to the Ziv O. paper, we have also crosschecked the top interactions from Li. P et al and show that all of their top five interactions are abundantly captured in our data. This again indicates that our data is of high quality. In contrast to the recent structure papers on Zika, which have performed proximity mapping of a single Zika virus (Ziv. O et al) and of two Zika strains (Li. P et al)⁴, we performed structure mapping of four different Zika strains, reflecting the diversity of Zika viruses in the original African strain, the Brazil strain, French Polynesia strain, and the South East Asian Singapore strain. We believe that our data contains a rich repository of information, not only for studying the similarities, but also the differences, between the strains.

"We believe that our data contains a rich repository of information": ABSOLUTELY! which is why it is crucial to properly present and deliver the load of information to the large community of scientists that work on such viruses. Especially since there is little doubt that the data are of high quality.

We thank the reviewer for his/her positive comments. All of the sequencing data has been deposited in GEO (GSE106483, publicly available as of 28/02/2019) and we have expanded our Supplementary Figures and information in Supplementary Data files to increase the accessibility of our data.

5. The data dealing with unfolding of the RNA genome in the cell, and the role of the ribosome and the action of helicases is underdeveloped. To a certain degree, this is obvious – we know that the viral genome is the template for infection processes in the cell that must unwind it (replication and translation) and we know that it exists at some point as a fully double-stranded intermediate; hence the conclusion that "the structured state we observe in side cells is a result of active unwinding by enzymes..." and that the viral replicase machinery could play a role not novel is a bit odd because we knew this...its settled virology. So, I'm left wondering: what did we learn here?

We thank the reviewer for his/her comments. The ability to map RNA structures inside cells in a high throughput manner is a fairly new technological development. As such, much remains to be learnt with regards to the structural dynamics of RNA structures in vivo versus in vitro and under different cellular states. RNA structures have been shown to be more open inside cells than in vitro, and a few groups have identified the ribosome as the major helicase that unwinds RNA structures inside cells^{5, 6}. As we also observe that longer range interactions in dengue and Zika genomes tend to be disrupted in cells versus in vitro⁷, we tested the hypothesis that the ribosome is also the major helicase unwinding viral structures. Surprisingly, unlike mRNAs which gained structure when ribosomes are inhibited, dengue RNAs do not show significant structural changes upon ribosome inhibition. This suggests that while the ribosome is the major helicase acting on mRNAs inside cells, it is not the major helicase that is acting on dengue and Zika. We agree with the reviewer that while this is a potentially

interesting observation, more work is needed to inhibit the helicase to dissect structural heterogeneity of viral structures, and that it is outside the scope of this manuscript. We have now removed the section on ribosome inhibition and dengue/Zika structure to streamline the manuscript.

I support the author's decision to remove this aspect from the current manuscript.

We thank the reviewer for his/her support.

6. Related to the above, infection by these RNA viruses is a very dynamic process in which the RNA plays several roles. In addition, these roles change over time and thus the amount of RNA in a given population (translating, replicating, packaging) also changes. In addition, these viruses generate large amount of several non-coding subgenomic RNAs that contain part of the genomic RNA sequence (I do not think this is even mentioned but it is a critical point!), hence there are at least 2-3 different RNA species that share sequence during infection. Each may have different structures and be doing different things. The authors do mention different structural states existing, but this is not fleshed out and thus I am left feeling like I don't really know much more about the reality of the structure of these RNAs in the cell than I did before. While the authors do not need to have all the answers, they need to talk about these data in light of these virological realities, otherwise it is superficial. This could be a paper in and of itself, if properly controlled and discussed.

We thank the reviewer for his/her comments. Due to short read sequencing, we are unable to distinguish between sfRNA from the full length virus RNA. We have now discussed the complexities of the different structural states in the discussion section of the manuscript.

This aspect makes it even more essential to discuss alternative structures in a completely separate manuscript. Currently, the probing data is a blur of several RNA species of various lengths, which may adopt different structures, as reviewer #3 points out. It is important to be able to sort out whether the alternative structures are associated to different states of the same RNA, or to different RNAs with a similar sequence.

We thank the reviewer for his/her comments, and agree that further analysis of alternative structures could serve as an additional manuscript.

We agree with the reviewer's comments that it would be highly desirable to clarify whether the alternative structures we observe are indicative of heterogeneity across a number of viruses or are the result of dynamic changes within a single RNA molecule, we are currently unable to differentiate these cases on the basis of the present data. However, we are currently working towards clarifying this important distinction and will gladly submit an additional manuscript as soon as we finish collecting additional data regarding this question and are confident to make a conclusive determination based on this data.

7. Overall, the discussion section leaves many mysterious and unanswered questions. I'll just say that two paragraphs is not nearly enough to cover all the caveats, various interpretations, integration with known virology and RNA structure, etc. given the density and amount of data presented.

We thank the reviewer for his/her comments. We have now deepened the analysis of our manuscript and expanded our discussion to better integrate our data with known virology and to address various interpretations of our data.

Although the discussion appears to now be of a sufficient length for one manuscript, the only way to expand it would be to continue it in... additional papers.

We thank the reviewer for his/her comments, which has made the manuscript for accessible to the readers.

Minor points

Other points:

1. Fig 1c+d: I think clustering what's <0.5 separately from what's >0.5 hides away information that would be best covered by a spectrum of three or more colors. I don't think that a reactivity of 0.49 for example would correspond to a paired residue, although here it would be black. And how many reactivities are in that 0.3-0.7 range anyway? Better to stick to the same scale as on Fig 2b.

We thank the reviewer for his/her comments. We have now removed discretization of the data and use the continuum of NAI-MaP reactivities directly in our analysis. We have updated Figure 1c, Figure 2b, Extended data 2c, 3b and 5a in the manuscript.

This point has been properly addressed.

2. lines 125-127: "As expected, regions that share similar structure patterns between the 8 viruses generally share higher sequence identity, consistent with the idea that sequence plays an important role in RNA structure". It's pretty obvious that sequence is important for structure formation so this statement comes off as odd.

We thank the reviewer for his comments- as structure similarity tends to trend with sequence similarity, we used increased structure similarity over sequence similarity to identify structurally conserved elements. We have re-worded this in the manuscript.

This point has been properly addressed.

3. line 146: Converting SHAPE reactivity to pseudo-energy terms has been shown by others to not be appropriate, unless that pseudo energy term is not regarded as an energy, but as a probabilistic inference. It's mostly useful in the context of such long RNAs when shown to support structures derived from other methods, such as sequence alignment, or probing using other probes (see Eddy SR, Annu Rev Biophys 43:433-456 (2014)).

We thank the reviewer for his/her comment. We calculated the distribution of SHAPE reactivity in our data for known 5' and 3' structural elements across all our systems and compared them to the analysis presented by Eddy SR, 2014 paper (Figure 6 in this rebuttal)¹. We observe distributions of reactivity for known and unknown bases congruent with what is presented by Eddy, namely that the most likely reactivity observed is low in case of paired and unpaired bases, but that the likelihood ratio for higher-reactivity bases indicates a high probability of unpairedness. We changed the wording of our manuscript to highlight that SHAPE reactivity is used for probabilistic inference of likely unpaired regions and added this analysis as Extended data 2f in the manuscript.

This point has been properly addressed.

4. Fig 2b: I see many regions with problems in there: see 69-88, 782-802, 2247-2275, 4090-4120, either they are paired and should not be shown as unpaired, or vice-versa. What's going on here?

We thank the reviewer for his/her comments. NAI-MaP reactivities that do not fit modelled structures could be due to: 1) the natural distribution of NAI-MaP reactivities in paired and unpaired bases; and/or 2) the fact that bases exist in alternative conformations in the virions (Figure 7 in this rebuttal).

These examples constitute further reasons for presenting and discussing alternative structures in more details!